# APOBEC mutagenesis is a common process in normal human small intestine

Yichen Wang [1], Philip S. Robinson [1,2], Tim H. H. Coorens [1,3], Luiza Moore [1,4], Henry Lee-Six [1], Ayesha Noorani[1], Mathijs A. Sanders[1], Hyunchul Jung[1], Riku Katainen[5], Robert Heuschkel[6], Roxanne Brunton-Sim[7], Robyn Weston[8], Debbie Read[8], Beverley Nobbs[8], Rebecca C. Fitzgerald [9], Kourosh Saeb-Parsy [10], Iñigo Martincorena [1], Peter J. Campbell [1], Simon Rushbrook[7,11], Matthias Zilbauer [2,6], Simon James Alexander Buczacki[12] & Michael R. Stratton [1] ✉

APOBEC mutational signatures SBS2 and SBS13 are common in many human cancer types. However, there is an incomplete understanding of its stimulus, when it occurs in the progression from normal to cancer cell and the APOBEC enzymes responsible. Here we whole-genome sequenced 342 microdissected normal epithelial crypts from the small intestines of 39 individuals and found that SBS2/SBS13 mutations were present in 17% of crypts, more frequent than most other normal tissues. Crypts with SBS2/SBS13 often had immediate crypt neighbors without SBS2/SBS13, suggesting that the underlying cause of SBS2/SBS13 is cell-intrinsic. APOBEC mutagenesis occurred in an episodic manner throughout the human lifespan, including in young children. *APOBEC1* mRNA levels were very high in the small intestine epithelium, but low in the large intestine epithelium and other tissues. The results suggest that the high levels of SBS2/SBS13 in the small intestine are collateral damage from APOBEC1 fulfilling its physiological function of editing *APOB* mRNA.

Somatic mutations are thought to accumulate in all cells. Although many cancer genomes have been comprehensively characterized[1–3], information on somatic mutations in normal cells has been limited. Recently, however, multiple technological advances[4,5] have enabled the detection of somatic mutations in many normal cell types, including blood[6], placenta[7], neurons, smooth muscle[5], cardiac muscle[8], epithelia of the liver[9], bronchus[10], endometrium[11], colorectum[12], skin[13], esophagus[14], bladder[15], pancreas, prostate, ureter, thyroid, visceral fat, adrenal gland and testis[8]. These studies have informed on the clonal structure of tissues, somatic mutation rates, mutational processes and the presence of driver mutations conferring selection in normal cells of healthy individuals, and those with a range of diseases.

The small intestine is the longest segment of the gastrointestinal tract and a major organ involved in the digestion and absorption of nutrients. Its epithelium is thought to be one of the most vigorously self-renewing tissues of adult mammals[16]. However, small intestine

[1]Cancer, Ageing and Somatic Mutation, Wellcome Sanger Institute, Hinxton, UK. [2]Department of Paediatrics, University of Cambridge, Cambridge, UK. [3]Broad Institute of MIT and Harvard, Cambridge, MA, USA. [4]Department of Pathology, Cambridge University Hospitals NHS Foundation Trust, Cambridge, UK. [5]Applied Tumor Genomics Research Program, Faculty of Medicine, University of Helsinki, Helsinki, Finland. [6]Department of Paediatric Gastroenterology, Hepatology and Nutrition, Addenbrooke's Hospital, Cambridge, UK. [7]Norfolk and Norwich University Hospital, Norwich, UK. [8]NIHR Clinical Research Network—East of England, Addenbrooke's Hospital, Cambridge, UK. [9]The Early Cancer Institute, Department of Oncology, University of Cambridge, Cambridge, UK. [10]Department of Surgery and Cambridge NIHR Biomedical Research Centre, Biomedical Campus, University of Cambridge, Cambridge, UK. [11]Norwich Medical School, University of East Anglia, Norwich, UK. [12]Nuffield Department of Surgical Sciences, Medical Sciences Division, University of Oxford, Oxford, UK. ✉e-mail: mrs@sanger.ac.uk

tumors constitute only ~4% of all gastrointestinal tumors[17]. Although a few normal small intestine crypts have been analyzed as parts of other studies[8,12,18], extensive sequencing of the normal small intestine epithelium has not thus far been conducted. To provide a comprehensive analysis of somatic mutations in the normal small intestine epithelium, we used the laser capture microdissection microscopy (LCM) and whole-genome sequenced 342 individual small intestine crypts from 39 individuals (173 duodenal from 22 individuals, 47 jejunal from 5 and 122 ileal from 16) aged between 4 and 82 years. Six had a history of celiac disease (gluten enteropathy). From each biopsy, we aimed to collect both spatially adjacent and separated crypts whenever possible. The mean sequencing coverage was 25 fold.

## Results

### The landscape of somatic mutation in normal human small intestinal crypts

The base of each small intestinal crypt is occupied by stem cells, and the descendants of a single recent ancestor stem cell comprise most cells in each crypt[19,20]. Therefore, isolation of single crypts provides relatively homogeneous clones of cells from which somatic mutations can be called. The distribution of variant allele frequencies (VAFs) around 0.5 in most crypts confirmed their monoclonality (Extended Data Fig. 1). From the distribution of VAFs, we estimated that the mean time to the most recent common ancestor (MRCA) of crypts ranges from 0.8–4.6 years (95% confidence interval, 0–10 years) across individuals, with a median of 2.6.

In total, we identified 787,109 unique single-base substitutions (SBS) and 51,256 small insertions and deletions (IDs). We fitted linear mixed-effects models to estimate the effect of age, biopsy location and disease condition on SBS and ID mutation burdens while controlling for within-patient correlation. On average per year, small intestine accumulated 51 SBS (95% confidence interval, 45–56) and 3.7 IDs (95% confidence interval, 3.1–4.4) per crypt in duodenum, 50 SBS (95% confidence interval, 42–59) and 2.6 IDs (95% confidence interval, 1.8–3.5) per crypt in jejunum, and 42 SBS (95% confidence interval, 35–48) and 2.3 IDs (95% confidence interval, 1.6–3.0) per crypt in ileum (Fig. 1a,b). Both SBS and ID rates are similar to normal colorectum[12], indicating that the low incidence rate of small bowel cancer is not due to lower mutation rates. For IDs, 1 bp IDs of A:T base pairs were the most common mutation categories and are thought to reflect polymerase slippage during DNA replication (Extended Data Fig. 2). Six crypts showed one copy number change (gain or loss). One crypt (PD45771b_lo0004) showed a chromosome 2 trisomy, while the remaining five were deletions ranging from 1 Mb to 7 Mb. Translocation and inversion events were detected in five crypts (Supplementary Table 1), and 96 retrotransposition events were identified in 65 small intestinal crypts (Supplementary Table 2). Hotspot putative cancer driver mutations in *FBXW7* (S582L), *ERBB2* (V842I) and *PIK3CA* (Q546E) were detected in one individual each. Seven heterozygous protein-truncating mutations were found in tumor suppressor genes including *RB1*, *FBXO11*, *FAT1*, *KMT2D*, *KMD6A*, *ACVR2A* and *ZFHX3* (Supplementary Table 3).

### Mutational signatures in normal human small intestinal crypts

Previous studies of somatic mutations have provided a set of reference mutational signatures[1,2,21]. By de novo extraction and decomposition of extracted signatures, we identified the following ten reference SBS signatures in normal small intestine epithelium: SBS1, SBS2, SBS5, SBS13, SBS17b, SBS18, SBS35, SBS40, SBS41 and SBS88 (Fig. 1c and Supplementary Table 4). Phylogenetic trees of small intestine crypt mutations were constructed for each individual and mutation burdens of each signature were estimated for each branch (Extended Data Fig. 3). SBS1 is characterized by N$\underline{C}$G>N$\underline{T}$G (mutated base underlined) substitutions and is due to spontaneous deamination of 5-methylcytosine. SBS5 is of unknown etiology but is likely of intrinsic origin. SBS1 and SBS5 are ubiquitous in human tissues and generally accumulate in a linear,

clock-like fashion throughout life[8,12,22]. Congruent with these general patterns, SBS1 and SBS5 were present in almost all small intestine crypts, and their mutation burdens correlated linearly with age ($r = 0.77$ for SBS1, $r = 0.90$ for SBS5; Extended Data Fig. 4). SBS18 is predominantly characterized by C>A substitutions, is proposed to be caused by DNA damage due to reactive oxygen species[14] and has been previously found in some normal human tissues, including the colorectal epithelium that has the highest SBS18 mutation rate other than the placenta[7,12]. On average, SBS18 contributed 12% of the total SBS burden of the small intestine, similar to that of the colorectum (13%)[6].

Seven signatures (SBS2, SBS13, SBS17b, SBS35, SBS40, SBS41 and SBS88) were present only in a subset of crypts and were thus considered to be 'sporadic'. SBS17b and SBS35 can be due to prior chemotherapy treatment with fluorouracil (5-FU) and platinum drugs, respectively. SBS17b was found in one individual (PD43853) previously treated with 5-FU, and SBS35 in two individuals (PD28690 and PD43853) previously treated with platinum drugs. In PD43853, SBS17b and SBS35 were extracted from clonal mutations (VAF > 0.4) in crypts sampled 5 months after commencement of chemotherapy, indicating that the time required for progeny of a single stem cell to colonize the whole crypt was less than 5 months. However, chemotherapy itself may influence stem cell dynamics and, therefore, this short period may not be representative of crypts in untreated individuals. SBS40 is a flat signature correlated with age, detected in ten individuals (PD37449, PD42835, PD43400, PD43403, PD43850, PD43851, PD45766, PD45770, PD46562 and PD46565). SBS41 is of unknown etiology and was present in three individuals (PD37449, PD46565 and PD46566). SBS88 was previously identified in subsets of colorectal crypts in a subset of individuals, is caused by the mutagenic agent colibactin produced by certain strains of *Escherichia coli* present in the colorectal microbiome[23] and usually appears to be generated during childhood[12]. Consistent with this pattern, SBS88 in the small intestine was present only in the earliest branches of phylogenetic trees constructed from somatic mutations. In PD37449, SBS88 constituted 52% of mutations in an ancestral branch and was not present in descendant branches, further refining the timing of colibactin exposure to a very early period of postgestational life, around or before 2 years based on SBS1 burden (Extended Data Fig. 3). Although the small intestine does not harbor the rich microbiome of the colon, all crypts with SBS88 were from the ileum, and it is conceivable that they had been exposed to colibactin through backwash from the colon.

### Frequency and burden of APOBEC mutagenesis

SBS2 and SBS13 are thought to be caused by activity of APOBEC cytidine deaminases (in this context, we subsequently use SBS2/SBS13 and APOBEC mutagenesis interchangeably). These signatures are characterized predominantly by C>T (SBS2) and C>G/C>A (SBS13) mutations at T$\underline{C}$N trinucleotides (mutated base underlined), usually occur together and are commonly found in many cancer types including bladder, breast, cervix, head and neck, esophageal squamous, lung squamous, lung adenocarcinoma, pancreas, stomach, thyroid and uterus[1,2,24]. SBS2/SBS13 have also been found in normal bronchus[10] and bladder epithelial cells[15] but have not been commonly detected in most normal tissues including the liver[9], endometrium[11], placenta[7], colon[12,25], stomach[8], skin[13], esophagus[14,26], neurons[5], cardiac muscle[8], smooth muscle[5], hematopoietic stem cells[6], pancreas, prostate, ureter, thyroid, visceral fat, adrenal gland and testis[8]. SBS2/SBS13 were observed in 22 out of 39 individuals, in 58/342 (17%) small intestine crypts, and on average contributed to 11% of the SBS burden in these crypts. This frequency of APOBEC mutagenesis in normal small intestinal epithelium is considerably higher than that observed in normal colorectal epithelium. To directly compare large intestine and small intestine epithelia, we combined our data with a previous dataset[25] of whole-genome sequences of normal colorectal crypts with mutations assigned to phylogenetic trees and ran a mutational

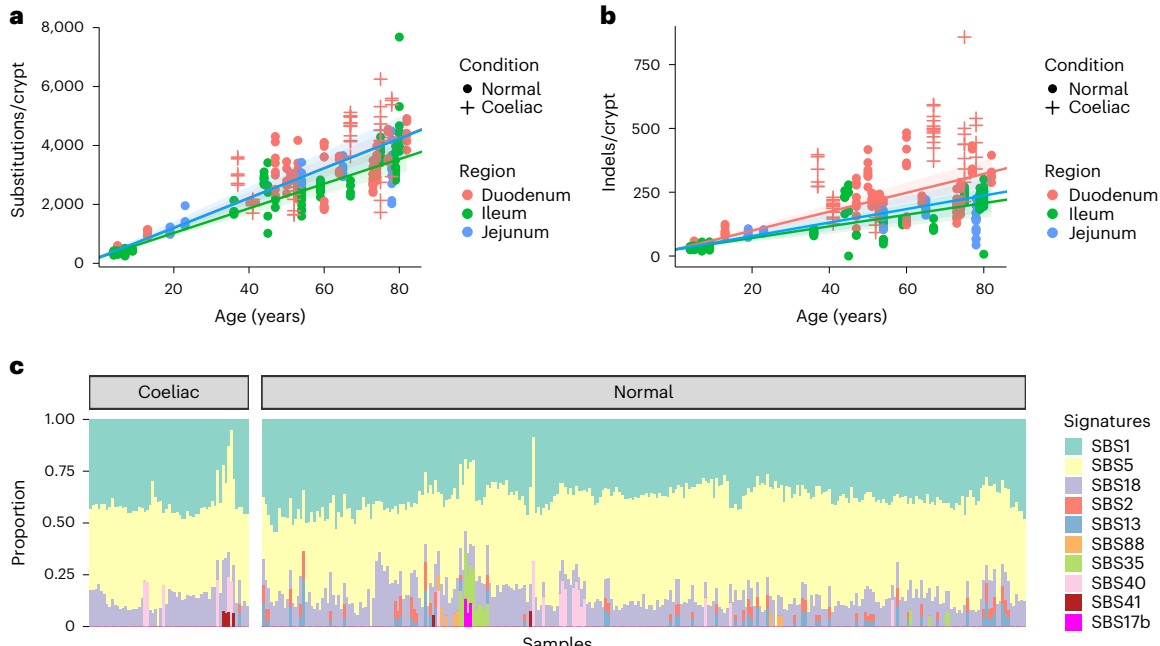

**Fig. 1 | Burdens and mutational signatures in normal human small intestinal crypts. a**, SBS burden versus age, showing regression lines for the three different sectors of the small intestine. Regression lines were estimated using linear mixed-effects models. Error bands represent 95% confidence interval for the fixed effect of age. Colors indicate biopsy regions, with orange, green and blue representing duodenum, ileum and jejunum, respectively. Shapes indicate whether the donor has a celiac history or not. Crosses indicate donors with a celiac history, and dots indicate donors without a celiac history. **b**, ID burden versus age, showing regression lines for the three different sectors of the small intestine. **c**, The proportion of mutations in each crypt attributed to each SBS mutational signature (arranged by ascending age). Signatures are color coded as indicated on the right.

signature extraction on all branches of the combined dataset. SBS2/SBS13 were found in 68 of 417 (16.3%) phylogenetic branches in small intestine, ~28-fold more frequently than the 6 of 1075 (0.6%) branches in colorectum ($P = 1.6 \times 10^{-35}$, chi-squared test). The proportion of all mutations that were SBS2/SBS13 was, however, only 8.5-fold higher in small intestine (2.1%) than in large intestine (0.24%) because of a very high mutation burden (>3,000) in a single large intestine crypt (Fig. 1c).

To investigate whether SBS2/SBS13 is also frequently present in small intestine cancers, we re-analyzed a whole-exome sequencing (WES) dataset of 71 small bowel adenocarcinomas[27] and the results showed a similar frequency (18%, 13 of 71) to normal small intestinal crypts. However, the median burden of SBS2/SBS13 from signature attribution in small bowel adenocarcinoma was ~7-fold higher than that observed in normal small intestine, suggesting that rates of APOBEC mutagenesis are accelerated during the process of neoplastic change and progression (Extended Data Fig. 4).

The high prevalence of SBS2/SBS13 and the ordered epithelial crypt structure of the small intestine offer a particular opportunity to investigate mechanisms of APOBEC mutagenesis. The proportions of APOBEC-positive crypts varied between individuals ($P = 5 \times 10^{-4}$, Fisher's exact test). These differences were not well-explained by age ($r = 0.19$ for SBS2 and age, $r = 0.18$ for SBS13 and age), by sector of the small intestine sampled or by a history of celiac disease (Extended Data Fig. 5). For example, in PD41851 (aged 80 years from the ileum), SBS2/SBS13 was found in 8/11 crypts, while in PD41853 (aged 79 years from the ileum) only 1/11 crypts showed these signatures (Fig. 2a,b). Thus, there appear to be differences between individuals in the extent of APOBEC mutagenesis.

**Spatial distribution of APOBEC-positive crypts**

The stimulus triggering SBS2/SBS13 mutagenesis is unknown. To investigate the possibility that APOBEC activity is triggered by extrinsic local microenvironmental factors that, in principle, might affect multiple crypts adjacent to each other, we examined the spatial relationships of crypts with SBS2/SBS13. Crypts with APOBEC mutagenesis often immediately neighbored crypts without APOBEC mutagenesis (Fig. 3). The results echo previous observations from normal bladder[15] and suggest that APOBEC mutagenesis is initiated or permitted by cell-intrinsic factors or, if not, by very highly localized extrinsic factors. APOBEC cytidine deaminases are thought to be involved in intrinsic immunity against retrotransposons[28–30]. However, no significant correlation between the number of retrotransposition events and SBS2/SBS13 mutation burden was found.

**APOBEC mutagenesis occurs episodically throughout life**

In vitro studies of human cancer cell lines have indicated that SBS2/SBS13 mutagenesis is episodic, occurring in bursts with extended periods of intervening silence[31]. To investigate whether APOBEC mutagenesis in normal small intestine cells in vivo is episodic, we examined crypt phylogenetic trees and found that APOBEC-positive branches usually had ancestral or descendant branches in which APOBEC mutagenesis was absent (Fig. 2d–f and Extended Data Fig. 3). The results, therefore, indicate that APOBEC mutagenesis is also episodic in vivo in normal cells and suggest that most adult small intestine cells have only experienced a single episode, or a small number of episodes, in the cell lineage from the fertilized egg spanning the lifetime of each individual.

One crypt from a 4-year-old exhibited SBS2/SBS13 demonstrating that APOBEC mutagenesis can occur early in life (Fig. 2c). Phylogenetic trees provided further information on the timing of APOBEC mutagenesis episodes. In PD43401 and PD43403, the absence of APOBEC signatures on branches ancestral to those in which APOBEC mutagenesis was found indicates that these episodes occurred after age 30 years (Fig. 2e,f). By contrast, in PD41852, APOBEC exposure on an early branch indicates the occurrence of an

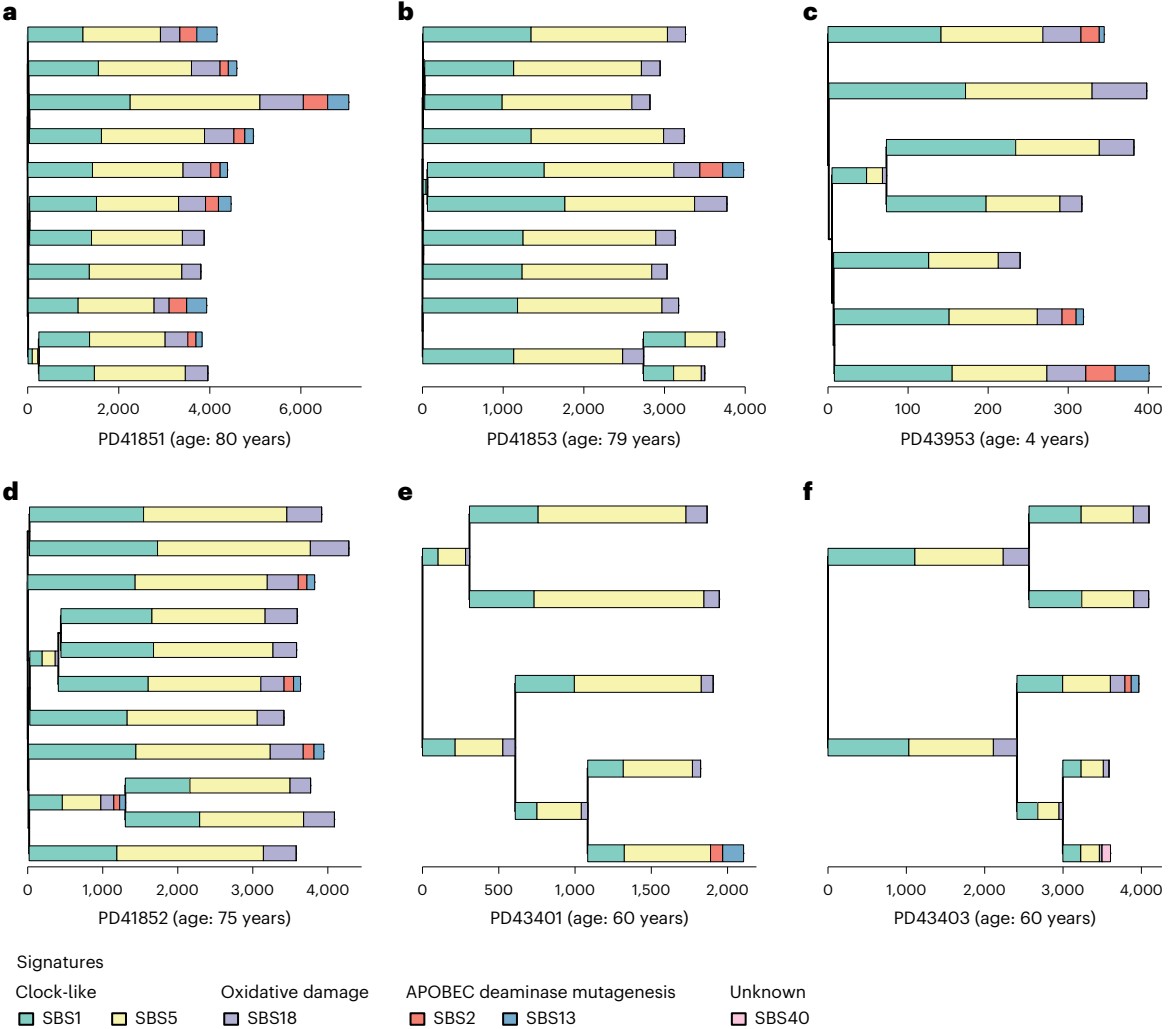

**Fig. 2 | APOBEC mutagenesis on phylogenetic trees.** Phylogenies of small intestine crypts with mutational signature annotation. Branch lengths correspond to SBS burdens. Signatures exposures are color coded below the trees. **a**, Phylogeny of PD41851, an individual with frequent APOBEC mutagenesis exhibiting SBS1, SBS5 and SBS18 with SBS2/SBS13 in 8 of 11 crypts. **b**, Phylogeny of PD41853, exhibiting SBS1, SBS5 and SBS18 with SBS2/SBS13 in 1 of 11 crypts. **c**, Phylogeny of PD43953, a 4-year-old child exhibiting SBS1, SBS5, SBS18 and SBS2/SBS13. **d**, Phylogeny of PD41852 with APOBEC mutagenesis detected before a crypt fission event at ~25 years of age. **e**, Phylogeny of PD43401 with APOBEC mutagenesis detected after a crypt fission event at ~30 years of age. **f**, Phylogeny of PD43403 with APOBEC mutagenesis detected after a crypt fission event at ~30 years of age.

episode before age 25 years (Fig. 2d). Together, the results suggest that APOBEC mutagenesis occurs in the small intestine throughout much of the lifespan.

## Kataegis clusters of APOBEC mutations

In addition to a more-or-less random genome-wide distribution, SBS2/SBS13 mutations in cancer are found in localized, high-density clusters referred to as 'kataegis'[32–34]. To investigate whether kataegis exists in normal small intestine, we applied a negative binominal test on all SBS mutations to identify mutation clusters spanning 10 to 10,000 base pairs and constructed 'rainfall' plots to visualize between-mutation distances. Eighty-seven kataegic clusters were identified, which, overall, showed fewer mutations than those in cancer genomes (Supplementary Table 5 and Fig. 4). Kataegis in cancers is often found around rearrangement breakpoints[1] but this was not the case in normal small intestine crypts. Among the clustered mutations identified, 88% (363 of 412, $P < 2.2 \times 10^{-22}$, chi-squared test) exhibited the characteristics of SBS2/SBS13, and 78% (321 of 412, $P < 2.2 \times 10^{-22}$, chi-squared test) co-occurred within crypts with >5% SBS2/SBS13 burden. Forty-eight percent of APOBEC-positive crypts (28 of 58,

$P = 9.7 \times 10^{-16}$, chi-squared test) had at least one kataegis focus, of which the earliest, by phylogenetic analysis, occurred before the age of ~30 years.

## Tissue-specific expression of *APOBEC1*

Multiple lines of evidence, including the sequence context of SBS2/SBS13 mutations in human cancers[35], engineered expression of APOBEC enzymes in model systems[35,36] and APOBEC gene knockouts[37] in human cancer cell lines, indicate that among the family of 11 APOBEC enzymes, APOBEC3A and, to a lesser extent, APOBEC3B are responsible for generating SBS2/SBS13. To investigate the underlying mechanism for the unusually high level of SBS2/SBS13 in small intestine epithelium and its sharp reduction on transition to the large intestine, we analyzed five independent datasets of publicly accessible bulk tissue[38] and single-cell transcriptome sequences[39–44]. This revealed that (1) *APOBEC1* is expressed at much higher levels in small intestine epithelium than large intestine epithelium (10- to 20-fold higher normalized TPM in bulk RNA sequencing (RNA-seq) data and ~10-fold higher relative read counts in single-cell RNA-seq (scRNA-seq) data of epithelium) and all other normal tissues (Fig. 5 and Supplementary Fig. 1), and (2) this

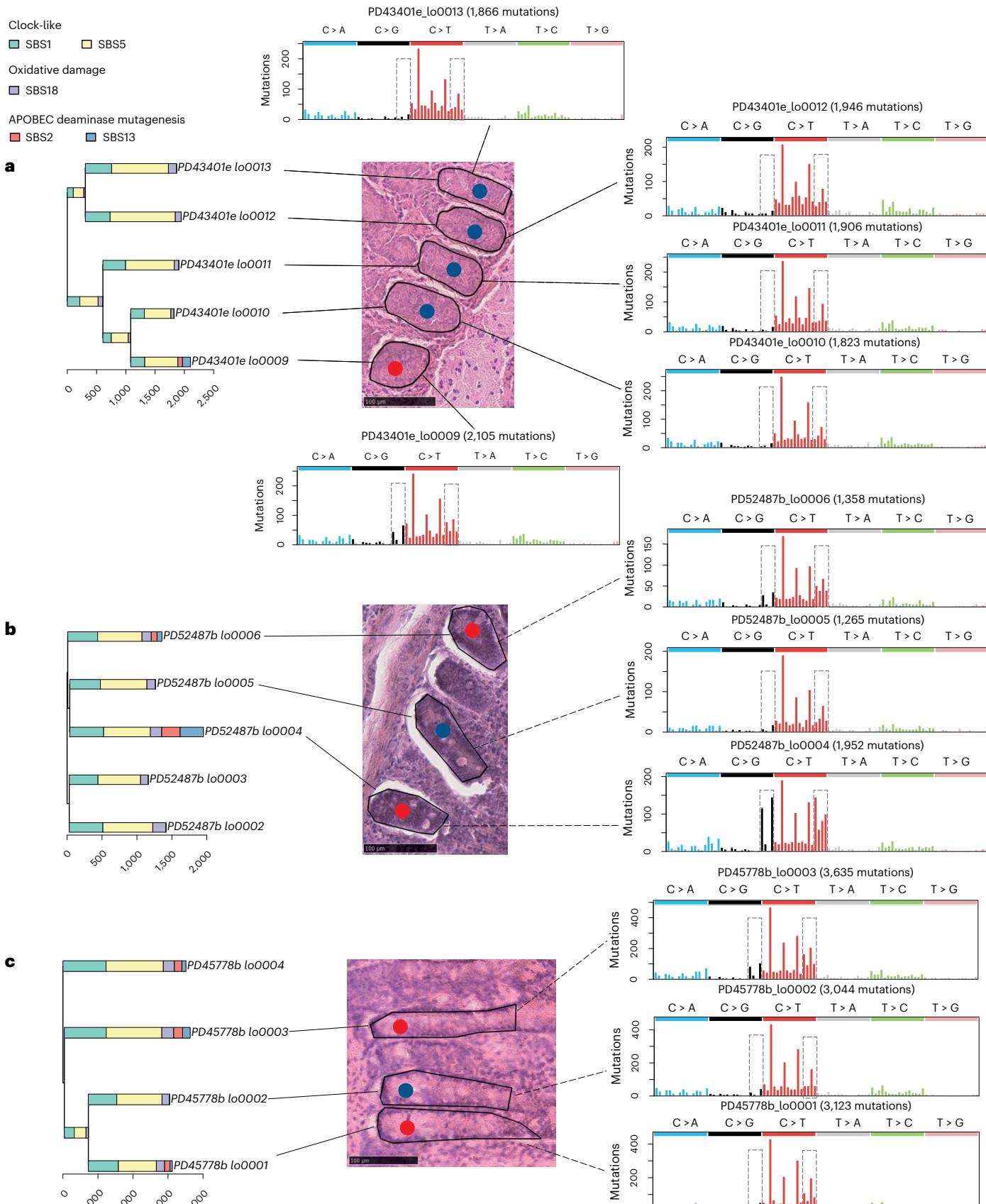

**Fig. 3 | Spatial distribution of APOBEC-positive crypts.** APOBEC-positive crypts and their surrounding crypts before microdissection with their SBS mutational spectrums. Signatures exposures are color coded on top left. Red dots, APOBEC-positive crypts. Blue dots, APOBEC-negative crypts that have been sequenced. Gray rectangles on the mutational spectra circle characteristic peaks of SBS2/SBS13. **a**, PD43401, this individual has one APOBEC-positive crypt but all the remaining crypts in the neighborhood are negative. **b**, PD52487, with an APOBEC-negative crypt (PD52487b_lo0005) between APOBEC-positive crypts. **c**, PD45778, an APOBEC-negative crypt (PD45778b_lo0002) between APOBEC-positive crypts.

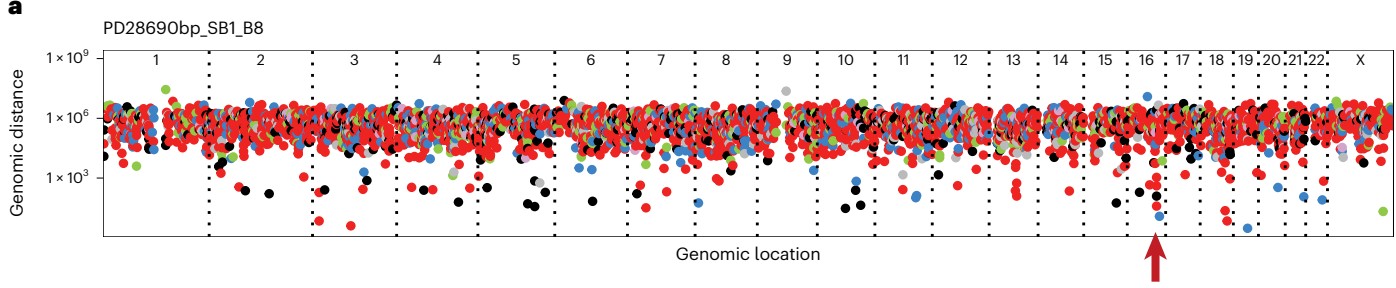

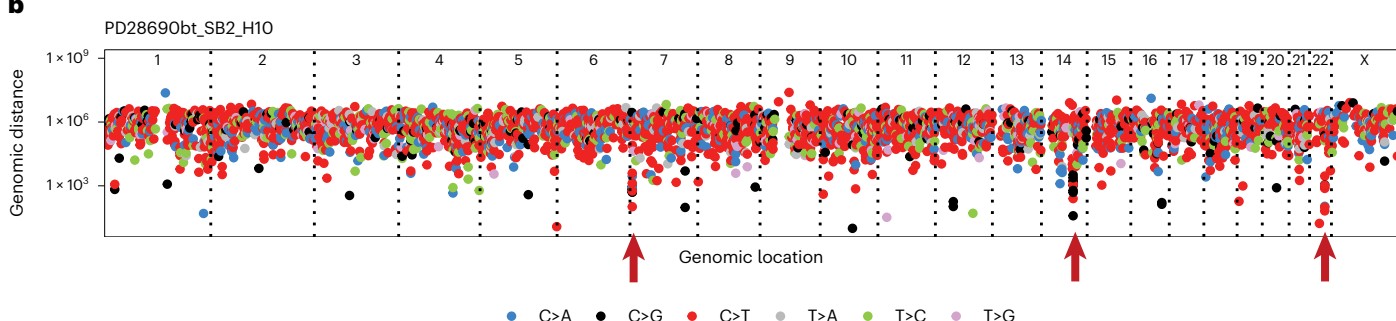

C>A    C>G    C>T    T>A    T>C    T>G

**Fig. 4 | Local mutation clusters (kataegis) in the normal small intestine.**
**a,b**, Two examples of crypts with kataegis. Types of SBSs are indicated by the color code below. Red arrowheads show the location of kataegis. **a**, Rainfall plot of a crypt from PD28690 showing a mutation cluster on chromosome 16. **b**, Rainfall plot of a crypt from PD28690 showing mutation clusters on chromosomes 7, 14 and 22.

difference in *APOBEC1* expression is even more pronounced between small and large intestine epithelial stem cells/transit amplifying cells in which the SBS2/SBS13 mutations found here must have been generated (20–40-fold higher relative read counts). Moreover, four of the five datasets indicate that (3) *APOBEC1* is expressed at higher levels than *APOBEC3A* and *APOBEC3B* in small intestine epithelium and, (4) in contrast to *APOBEC1*, *APOBEC3A* and *APOBEC3B* show lower expression in small intestine than in large intestine epithelium (Table 1).

APOBEC1 has a known physiological function, mediating C>U editing at position 6,666 in apolipoprotein B (*APOB*) mRNA to generate a truncated APOB protein with specific lipid trafficking capabilities critical to normal lipid absorption and distribution by the small intestine[45,46]. In addition to its RNA editing capability, in experimental systems, APOBEC1 generates C>T mutations in DNA with a sequence context similar to that of APOBEC3A[47–50], and consistent with that observed here in small intestine SBS2/SBS13 mutations (Extended Data Fig. 6 and Supplementary Fig. 2). Thus, it is plausible that the 10–40-fold higher level of *APOBEC1* mRNA expression in small compared to the large intestine epithelium is responsible for the ~28-fold higher SBS2/SBS13 frequency in small compared to the large intestine epithelium, and the even greater differences compared to most other normal tissues.

**The impact of celiac disease on somatic mutations**
In celiac disease (gluten enteropathy), the immune system response to gluten is directed at the small intestine epithelium, resulting in villous atrophy and elevated risks of lymphoma and small intestine carcinoma. Although the number of affected individuals is small and estimates correspondingly uncertain, celiac disease increased the SBS1 burden by 4.8 mutations per year (95% confidence interval, 0.4–9.2; $P = 0.034$ by $t$-test) and indel burden by 1.6 IDs per year (95% confidence interval, 0.3–2.9; $P = 0.017$ by $t$-test), indicative of 1.43-fold increased small indel and 1.24-fold increased SBS1 mutation rates (in these analyses, we used age to estimate mutation rates and therefore the mutation rates during active disease periods could be

considerably higher) (Extended Data Fig. 7). However, the total mutation rate, driver mutation rate, complement of mutational signatures and rates of mutational signatures other than SBS1 were not detectably affected. The SBS1 mutation rate has previously been correlated with cell division rates[22], suggesting that there is accelerated crypt stem cell proliferation in celiac disease.

## Discussion
This study shows that the total somatic mutation rates of small intestine stem cells are similar to those of the colorectum, confirming previous results[12,25]. Thus, the markedly lower cancer incidence in the small bowel compared to the large bowel is not explained by lower mutation burdens in adult cells.

APOBEC mutagenesis is found frequently in small intestine epithelium compared to the large intestine epithelium and most other cell types thus far investigated, and the frequency of crypts showing APOBEC mutagenesis differs between individuals. The nature of the stimulus triggering APOBEC mutagenesis remains elusive but the results suggest that it is controlled by cell-intrinsic factors, is episodic and can initiate APOBEC mutagenesis during the whole human lifespan, albeit on few occasions in each cell lineage from fertilized egg to normal adult small intestine cell.

APOBEC1 has rarely been considered[51,52] as a contributor to SBS2/SBS13 mutation burden in cancer or normal tissues because of its small intestine-specific expression profile. However, the association between the 10- and 40-fold differences in *APOBEC1* mRNA expression levels and the ~28-fold difference in SBS2/SBS13 frequency comparing small and large intestine epithelia provides strong circumstantial evidence that APOBEC1 is responsible for the high SBS2/SBS13 mutation levels in normal small intestine. A definitive examination of this hypothesis would be provided by *APOBEC1* knockout in organoids derived from normal small intestine epithelium, although if SBS2/SBS13 mutation episodes are as infrequent in vitro as in vivo, these might be daunting experiments to conduct. If correct, however, this indicates that APOBEC1, in addition to APOBEC3A and APOBEC3B, can contribute to SBS2/SBS13

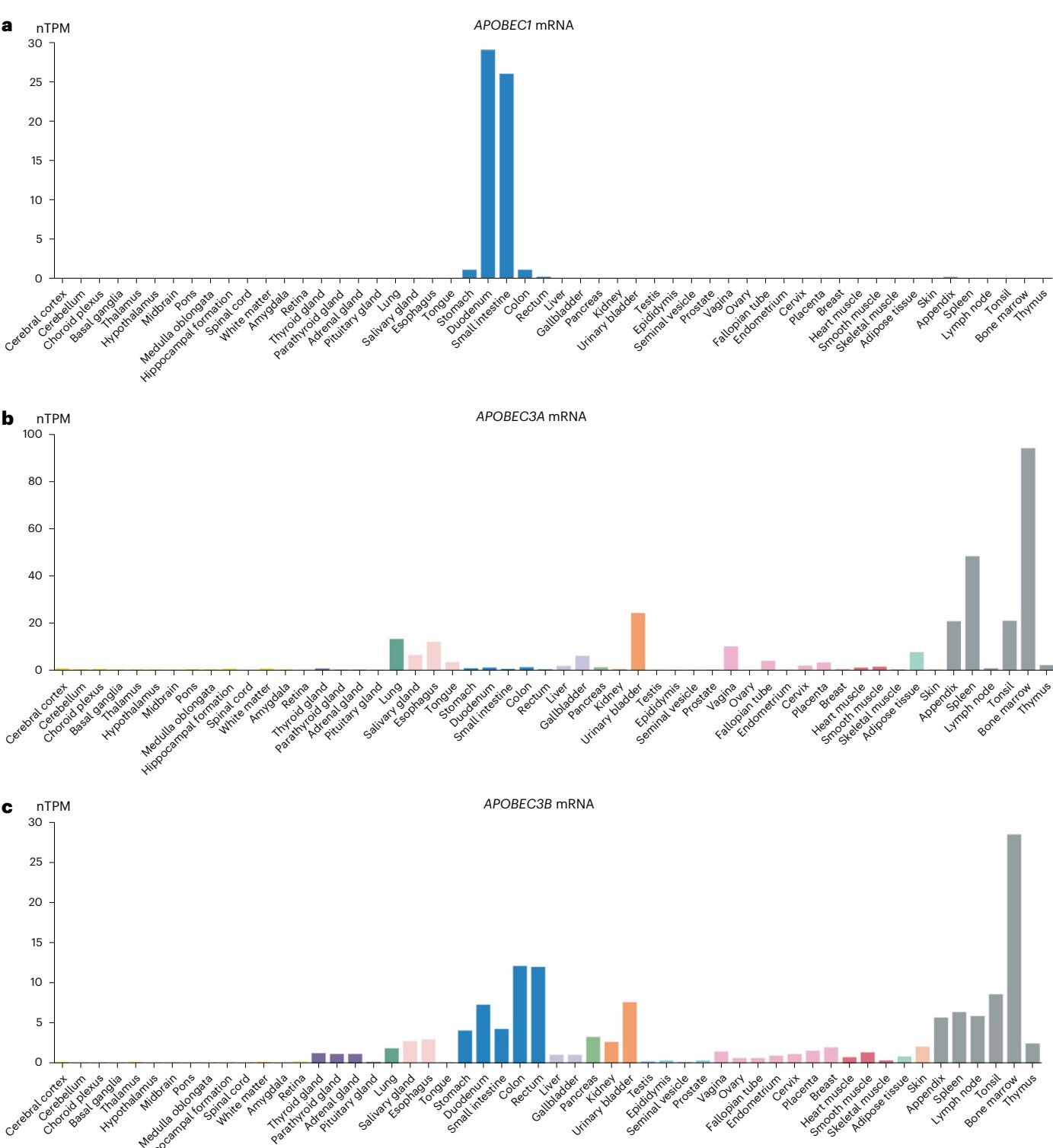

**Fig. 5 | *APOBEC1*/*APOBEC3A*/*APOBEC3B* expression across normal tissues.**
Bulk tissue gene APOBEC expression from The HPA project consensus dataset[38].
Image credit: HPA (v21.proteinatlas.org). **a**, *APOBEC1* bulk tissue gene expression
(https://v21.proteinatlas.org/ENSG00000111701-APOBEC1/tissue). Duodenum
nTPM = 29, small intestine nTPM = 26 and colon nTPM = 1.1. **b**, *APOBEC3A* bulk
tissue gene expression (https://v21.proteinatlas.org/ENSG00000128383-
APOBEC3A/tissue). Duodenum nTPM = 1.4, small intestine nTPM = 0.8 and colon
nTPM = 1.5. **c**, *APOBEC3B* (https://v21.proteinatlas.org/ENSG00000179750-
APOBEC3B/tissue) bulk tissue gene expression. Duodenum nTPM = 7.3, small
intestine nTPM = 4.3 and colon nTPM = 12.1.

mutations in human cells, and, therefore, that APOBEC1 performs
both RNA editing and DNA editing in normal small intestine. Given
the established physiological function of APOBEC1 in editing *APOB*
mRNA, it also leads to the conjecture that either APOBEC1 has multiple

physiological functions, some mediated by RNA editing and others by
DNA editing, or that the DNA editing leading to SBS2/SBS13 is simply
collateral damage arising as a result of the high levels of APOBEC1
required to serve its role in *APOB* mRNA editing. The observation that

**Table 1 | *APOBEC1/APOBEC3A/APOBEC3B* expression in small and large intestines across bulk and single-cell RNA-seq datasets**

| Dataset | Data type | Gene | Small intestine | Large intestine | Small intestine/large intestine | Adj *P* value |
|---|---|---|---|---|---|---|
| The HPA[38] project | Bulk RNA-seq (nTPM) | *APOBEC1* | 26 | 1.1 | 23.6 | – |
| | | *APOBEC3A* | 0.8 | 1.5 | 0.53 | – |
| | | *APOBEC3B* | 4.3 | 12.1 | 0.36 | – |
| The GTEx project | Bulk RNA-seq (nTPM) | *APOBEC1* | 11.9 | 1.1 | 10.8 | – |
| | | *APOBEC3A* | 0.8 | 1.4 | 0.57 | – |
| | | *APOBEC3B* | 2.1 | 6 | 0.35 | – |
| Gut Cell Survey[39–41] epithelial | scRNA-seq (relative read count) | *APOBEC1* | 0.27 | 0.028 | 9.72*** | 0 |
| | | *APOBEC3A* | 0.016 | 0.082 | 0.20*** | $5.0×10^{-91}$ |
| | | *APOBEC3B* | 0.016 | 0.053 | 0.31*** | $9.25×10^{-205}$ |
| Gut Cell Survey[39–41] stem cells | scRNA-seq (relative read count) | *APOBEC1* | 0.024 | 0.00058 | 41.6*** | $9.0×10^{-4}$ |
| | | *APOBEC3A* | 0.0026 | 0.015 | 0.17 | 1 |
| | | *APOBEC3B* | 0.022 | 0.036 | 0.62 | 0.73 |
| Tabula Sapiens[42] epithelial | scRNA-seq (relative read count) | *APOBEC1* | 0.12 | 0.012 | 9.29*** | $2.3×10^{-79}$ |
| | | *APOBEC3A* | 0.17 | 0.021 | 8.29*** | $5.3×10^{-55}$ |
| | | *APOBEC3B* | 0.030 | 0.0086 | 3.48*** | $2.2×10^{-13}$ |
| Tabula Sapiens[42] stem/amplifying cells | scRNA-seq (relative read count) | *APOBEC1* | 0.022 | 0 | Infinity | 1 |
| | | *APOBEC3A* | 0.080 | 0.025 | 3.16 | 0.38 |
| | | *APOBEC3B* | 0.056 | 0.038 | 1.50 | 1 |
| HPA collection[43,44] epithelial | scRNA-seq (relative read count) | *APOBEC1* | 0.62 | 0.062 | 10.0*** | $7.3×10^{-246}$ |
| | | *APOBEC3A* | 0.021 | 0.22 | 0.095 | 0.86 |
| | | *APOBEC3B* | 0.0047 | 0.059 | 0.080*** | $7.0×10^{-9}$ |
| HPA collection[43,44] undifferentiated cells | scRNA-seq (relative read count) | *APOBEC1* | 0.11 | 0.0047 | 22.3*** | $2.4×10^{-6}$ |
| | | *APOBEC3A* | 0.0059 | 0.053 | 0.11 | 1 |
| | | *APOBEC3B* | 0.012 | 0.028 | 0.45 | 1 |

Relative read count was calculated under a scale factor of $10^4$ and averaged across all cells. All single-cell data were obtained by 10X Genomics droplet-based sequencing. Gut Cell Survey—73,023/47,020 cells and 6,952/1,065 stem cells from small/large intestine from small/large intestine. Tabula Sapiens—3,112/5,790 cells and 259/318 stem and amplifying cells from small/large intestine. Human Protein Atlas (HPA) collection—6,167/11,167 cells and 1,290/2,104 undifferentiated cells from small/large intestine. *P* values were calculated from the coefficient two-tailed *t*-test of negative binomial regression models and were corrected for multiple testing using Benjamini–Hochberg method. Asterisks on the ratio indicate statistical significance. ***: Adjusted *P* value <0.001.

there are few episodes of APOBEC mutagenesis during the lifetime of an individual suggests that while APOBEC enzyme expression is necessary, it is not sufficient to generate SBS2 and SBS13 and that further, likely stochastic events are required.

## Online content

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

## Methods

### Ethics and overview

This study complies with all relevant ethical regulations and was approved by National Research Ethics Service Committee East of England (PD28690, PD37449, PD34200 and PD37266), London-Surrey Research Ethics Committee (PD43850, PD43851, PD52486 and PD52487), Wales Research Ethics Committee 7 (PD41851, PD41852, PD41853, PD42833, PD42834, PD42835 and PD43853), National Research Ethics Service Committee East of England−Cambridge East (PD43400, PD43401, PD43402 and PD43403), National Research Ethics Service Committee East of England−Cambridge South (PD43949, PD43950, PD43951, PD43952, PD43953 and PD43954), National Research Ethics Service Committee North West−Haydock (PD45766, PD45767, PD45769, PD45770, PD45771, PD45773, PD45776 and PD45778) and East of Scotland Research Ethics Service (PD46562, PD46563, PD46565, PD46566, PD46568 and PD46573). We obtained healthy and celiac small intestinal biopsies from 39 individuals (22 females and 17 males aged between 4 and 82 years from the United Kingdom). Among them, six individuals (two females and four males) had a history of celiac disease (gluten enteropathy). Participants were recruited from eight cohorts, including organ donors (providing duodenum, jejunum and ileum samples), patients who have undergone endoscopy (providing duodenum samples) and patients with colorectal cancer who have had surgical removal for part of their colon (including part of the ileum). Written, informed consents for all participants were obtained and participants were not compensated. A summary of age, sex and cohort is provided in Supplementary Table 6.

The first cohort consists of samples collected in a previous study[8] from a 78-year-old man (PD28690), a 54-year-old woman (PD43850) and a 47-year-old man (PD43851) in warm autopsies within 6 h of death. PD28690 was a nonsmoker who died of a metastatic esophageal adenocarcinoma for which he had received a short course of palliative chemotherapy (5–6 weeks of oxaliplatin, 7 weeks before his death). The samples were collected in line with the protocols approved by the National Research Ethics Service Committee East of England with United Kingdom Research Ethics Committee (REC) reference 13/EE/0043. The other two individuals died of causes not related to cancer. The use of these tissues was approved by the London-Surrey Research Ethics Committee (REC reference 17/LO/1801). Samples in the second cohort were collected in a previous study[12] from organ donors (two females and one male) at age 36–67 years from whom small-intestinal biopsies were taken at the time of organ donation (approved by the National Research Ethics Service Committee East of England, REC reference 15/EE/0152). The third cohort represents four female and three male patients aged 38–80 who underwent surgical resection (approved by Wales REC 7 with REC reference 15/WA/0131). The fourth represents four female patients aged 53–77 years who had endoscopy (approved by NRES Committee East of England−Cambridge East with REC reference 08/h0304/85+5). The fifth cohort comprised children aged between 4 and 13 years (four females and two males) who had endoscopy (approved by NRES Committee East of England−Cambridge South with REC reference 17/EE/0265). The sixth cohort comprised six male and two female patients aged 50–82 years with surgical resection (approved by NRES Committee North West−Haydock with REC reference 20/NW/0001). The seventh cohort comprised two female and four male patients aged between 41 and 78 years with celiac conditions, and samples were collected through endoscopy (approved by East of Scotland Research Ethics Service with REC reference 18/ES/0133). Celiac history information is provided in Supplementary Table 7. The last cohort is from AMSBio (commercial supplier), samples for donors PD52486 (19, female) and PD52487 (23, female) were obtained at autopsy from individuals who had died of causes not related to cancer (approved by the London-Surrey Research Ethics Committee with REC reference 17/LO/1801).

### Statistics and reproducibility

No statistical method was used to predetermine the sample size. The sample size was determined by the availability of tissue and the cost of the experiment. The experiments were not randomized. The Investigators were not blinded to allocation during experiments and outcome assessment.

All small intestinal crypts were included in the analysis, except when modeling mutation rates, samples were excluded with <15-fold coverage and from one individual (PD43853) with a substantial number of chemotherapy-induced mutations. In addition, two biopsies (PD43851j_P52_DDM_E2, PD46565c_lo0009) have a mutational landscape dominant by SBS5/40 and with lower mutation burden, which is distinct from the mutational landscape of normal small intestinal crypts, and similar to that of Brunner's glands, despite their crypt-like appearance under microscopy inspection. These two samples were kept and reported for the comprehensiveness and transparency of this dataset but were not included in the statistical modeling of mutation rates (leaving 306 crypts). When modeling clonal dynamics, crypts from celiac patients (PD46562, PD46563, PD46565, PD46566, PD46568 and PD46573), children (PD43949, PD43950, PD43951, PD43952, PD43953 and PD43954) and the patient with chemotherapy mutations (PD28690 and PD43853) were excluded, because these crypts might have a different number of cells and nonconstant mutation rate.

### Laser-microdissection and low-input whole-genome sequencing

We followed the standard protocol established at Wellcome Sanger Institute for tissue processing, laser-microdissection, low-input library generation and variant calling[4]. Fresh frozen biopsies were fixed, embedded, sectioned and stained before library preparation. PAXgene FIX Kit (PreAnalytiX, 765312) was used for fixation. Subsequent paraffin embedding was applied for higher quality morphology of the tissue. Biopsies were sectioned (10–20 mm), fixed to 4 mm PEN membrane slides (11600288, Leica) and stained with hematoxylin and eosin. Crypts were isolated using LCM (LMD7000, Leica) and collected in separate wells of a 96-well plate. Collected samples were lysed using ARCTURUS PicoPure DNA extraction kit (Applied Biosystems) according to the manufacturer's instructions.

DNA library concentration was measured following library preparation and used to guide the choice of samples subject to DNA sequencing. The minimum library concentration was 5 ng μl$^{-1}$, and libraries with >15 ng μl$^{-1}$ were preferably picked. Paired-end sequencing reads (150 bp) were generated on Illumina NovaSeq platform and were aimed to reach a coverage of ~30x. Sequences were aligned to the human reference genome (NCBI build37) using BWA-MEM[53] (versions 0.7.12-r1039 and 0.7.17).

### Single-base-substitution calling and filtering

We used the Cancer Variants through Expectation Maximization (CaVEMan) algorithm[54] (versions 1.11.2, 1.14.1 and 1.15.1) to call single-base somatic substitutions by performing variant calling against an in silico human reference genome or a matched polyclonal sample from the same individual. Then a series of postprocessing filters were applied as previously described[4,7,10,12,18,25]—the first filter removed mapping artifacts associated with BWA-MEM as follows: the median alignment score of reads supporting a mutation should be greater than or equal to 140, and fewer than half of these reads should be clipped. The second filter was applied to remove artifacts that are associated with the LCM library preparation, the code of the first and second filters can be found at https://github.com/MathijsSanders/SangerLCMFiltering.

The third filter was applied to remove germline variants, for which we fitted a binomial distribution to test the global variant allele frequency at each variant site across all samples from one patient. Germline mutations will be present at global variant allele frequency ~0.5 (heterozygous) or 1 (homozygous), therefore we used a one-sided

exact binomial test, with the null hypothesis that these variants were drawn from a binomial distribution with a success probability $P = 0.5$ ($P = 0.95$ for sex chromosomes in males). The alternative hypothesis was that these variants were drawn from distributions with $P < 0.5$ (or $P < 0.95$). The resulting $P$ values were corrected for multiple testing with the Benjamini–Hochberg method and a cut-off was set at $q < 10^{-5}$. Variants with $q > 10^{-5}$ were classified as germline, and for the remaining variants, the null hypothesis could be rejected and therefore they were classified as somatic.

Finally, a beta-binomial filter was applied to filter out the remaining artifacts. These artifacts are often present at similarly low frequencies across samples, while true somatic variants will be present at a high VAF in some samples but absent in others. Therefore, we calculate the maximum likelihood of the overdispersion parameter (rho) for beta-binomial distribution of each variant. Any variant with an estimated rho smaller than 0.1 was filtered out. This filter is adapted from the Shearwater variant caller[55]. The code for the last two filters can be found at https://github.com/TimCoorens/Unmatched_NormSeq.

### Indel calling
Indels were called using Pindel[56] (cgpPindel versions 2.2.2 and 3.3.0) and similar filtering strategies as single-base-substitution filtering were applied. After passing the first two filters for mapping and LCM artifacts, variants that passed possessed a minimum quality score of ≥300 at positions covered by at least 15 reads and were subject to the same binomial and beta-binomial filters.

### Stem cell dynamics
We used approximate Bayesian computation to estimate the time to the MRCA stem cell of the crypt. We ran 50,000 simulations for each crypt using a uniform prior for the number of stem cells and stem cell replacement rate, under a constant mutation rate and estimated full crypts of 1,000 cells. Variant allele frequency distributions of the simulated crypts were compared to observed data, and the likelihood was estimated based on distance. Time to the MRCA was estimated using the best 1% simulation.

### Mutation rates
For comparison of mutation rates, we calculated a sensitivity score to represent the possibility of detecting a variant with a given sequence coverage, variant allele frequency and algorithm settings[7]. We ran 100,000 simulations and estimated the sensitivity score as the frequency of observing a mutant read at least four mutant reads for SNVs or five for indels (the minimum depth requirement for CaVEMan and Pindel calls, respectively) in each simulated run. Each simulation was a set of Bernoulli tests with a success probability equal to the median VAF of the sample, and the number of tests was drawn from a Poisson-distributed depth given the mean coverage. After dividing the mutation count by the sensitivity score, the adjusted counts reflect the rate of mutation in the sample.

We then fitted linear mixed-effects models using nlme R package (version 3.1–148) to estimate the contribution of age, biopsy location and disease condition to mutation burden. Because biopsies from the same individual are not independent, we controlled this within-patient correlation by estimating a random effect for each patient. ANOVA tests between models were used to test whether biopsy regions and disease conditions affect fixed and random effects. For SBS2/SBS13, we also tested whether retrotransposition affects the mutation burden. The code for this analysis can be found at https://github.com/YichenWang1/small_bowel/tree/main/Mutational_burden.

### Copy number variation, structural variants and retrotransposon events
Copy number variations were called using two independent software ASCAT (v4.0.1, v4.1.2 and v4.5.0)[57] and Battenberg (v3.5.3)[32] and detected via the breakpoints predicted by BRASS (v6.1.2 and v6.3.4)[11]. Structural variants were detected by BRASS and GRIDSS (v2.13.1)[58,59]. Intrachromosomal variants smaller than 1 MB, as well as anything in the matched normal, were filtered out. All copy number changes and structural variants were then validated by visual inspection in the genome browser JBrowse (1.15.2)[60]. Retrotransposition events were called by GRIDSS.

### Detection of driver mutations
To identify possible driver mutations, we overlapped filtered somatic mutations with a known list of genes under positive selection in human cancers[61]. Then the mutations were annotated using the cBioPortal MutationMapper cancer hotspot mutation database v5.1.7 (http://www.cbioportal.org/mutation_mapper)[62,63]. Mutations in known hotspots, as well as protein-truncating variants in putative tumor suppressor genes, were reported.

### Phylogeny reconstruction
Phylogenetic trees of small intestinal crypts were generated from the filtered substitutions using a maximum parsimony algorithm (MPBoot v1.1.0)[64]. Substitutions were mapped onto tree branches using a maximum likelihood approach and visualized using ggtree (v3.3.1)[65–67] and ape (v5.6.1)[68].

To estimate the age interval during which specific mutations happened, we used the time of the next crypt fission event as upper limits, and the time of the previous crypt fission event as lower limits. To estimate the time of crypt fission, we built a linear mixed-effects model for the clock-like signature SBS1 and estimated the age using the total SBS1 burden accumulated so far and SBS1 mutation rate for that individual.

### Identifying kataegis
Kataegis was determined based on a negative binomial test described in previous studies[24,34,35] and visualized using MutationalPatterns package (v3.0.1)[69]. Mutations within ten bases were treated as a single mutagenic event and mutation clusters were identified as consecutive mutations with any pair of adjacent mutations less than 10 kb. The $P$ value is calculated as follows:

$$P = \sum_{0}^{k} \binom{k+r-1}{r-1}(1-p)^k p^r$$

where $r$ is the number of mutations −1 in the cluster (number of successes), $k$ is the number of unmutated bases spanned by a group (number of failures) and $p$ is the mutation rate of the individual (probability of success). Mutation clusters with adjusted $P < 10^{-4}$ after Bonferroni correction were classified as kataegis foci. Output results were manually inspected to avoid counting shared mutation clusters multiple times.

### Mutational signature extraction and fitting
To explore possible undiscovered mutational signatures in small intestine, we first used a hierarchical Dirichlet process (HDP v0.1.5)[70] without priors to extract mutational signatures (Supplementary Fig. 3). Before running HDP, we assigned mutations to branches on the phylogenetic tree. This avoided counting the same mutation within one patient multiple times. To avoid overfitting, we only kept branches with >50 mutations as input. The HDP was run in 10 independent chains for 12,000 iterations and with a burn-in of 20,000.

For identified SBS signatures, signatures with ≥0.9 cosine similarity with the reference were considered the same signatures. For the remaining signatures, we first ran HDP with all known PCAWG reference signatures[2,21] as priors and kept the extracted signatures as a shortlist of candidate reference signatures (Supplementary Fig. 4). Then expectation maximization[7,8,18] was used to deconvolute the remaining signatures into the shortlisted reference signatures. We ran

a second round of expectation maximization that only kept reference signatures with >10% contributions for each HDP signature to reduce overfitting. HDP signature 1 was also deconvoluted in the same way despite its >0.9 cosine similarity with SBS1 because its spectrum clearly showed residues that are not from SBS1. At last, we found every HDP signature could be reconstructed to a spectrum >0.8 cosine similarity with the original using these shortlisted reference signatures, therefore we assumed no new signature was detected in this dataset. The final SBS mutational signatures permitted in each individual were the corresponding deconvoluted reference signatures for HDP components that contributed to at least 5% of mutations in at least one branch (with branch length >200) of the individual phylogenetic tree. The final SBS mutational signatures for each crypt/branch were the reference signatures that had >5% contribution to the total burden of the crypt/branch, and the final proportion of reference signatures was estimated using sigfit (v2.0)[71]. The code for this analysis can be found at https://github.com/YichenWang1/small_bowel/tree/main/Signatures.

### Comparison with the small bowel adenocarcinoma WES dataset

Mutational signatures in small bowel cancer samples were extracted in the same way as the normal crypts. Samples where the two APOBEC signatures SBS2/SBS13 have at least a 5% contribution to the mutation burden were classified as APOBEC-positive. Because exomes constitute ~2% of the whole genome, the number of APOBEC mutations in the cancer WES dataset was multiplied by 50, to enable a direct comparison of APOBEC mutagenesis burdens between cancer and normal.

### APOBEC mutation context analysis

We generated a mutation matrix with extended context and ran HDP de novo signature extraction to get an APOBEC signature with extended context. Sequence frequency plot was generated using WebLogo v2.8.2 (ref. [72]; https://weblogo.berkeley.edu/logo.cgi) from the spectrum of the extracted APOBEC signature. Sequence context frequencies were extracted using P-MACD[24] to calculate enrichment scores. APOBEC3A and APOBEC3B signatures are distinguishable by looking at the −2 position of the TCA motifs (position 0 is the mutated cytosine)[35]. APOBEC3A hypermutations are enriched in pyrimidines (Y) rather than purines (R) at position −2, while APOBEC3B hypermutations are enriched in purines instead of pyrimidines. Results of YTCA enrichment ($P = 3.0 \times 10^{-96}$, Fisher's exact test) instead of RTCA ($P = 1$, Fisher's exact test) exclude APOBEC3B as the major contributing enzyme in small intestine, while APOBEC1/3A could not be excluded.

### Transcriptomic analysis

We used bulk RNA-seq data across tissue from the genotype tissue expression (GTEx) project and the HPA project (v21.proteinatlas.org)[38] as below:

*APOBEC1*: https://v21.proteinatlas.org/ENSG00000111701-APOBEC1/tissue

*APOBEC3A*: https://v21.proteinatlas.org/ENSG00000128383-APOBEC3A/tissue

*APOBEC3B*: https://v21.proteinatlas.org/ENSG00000179750-APOBEC3B/tissue

The data are based on The Human Protein Atlas version 21.1 and Ensembl version 103.38. For bulk RNA-seq data, normalized expression (nTPM, normalized protein-coding transcripts per million), corresponding to mean values of the different individual samples from each tissue, was generated from the HPA normalization pipeline. Single-cell RNA-seq datasets were obtained from Gut Cell Survey[39–41] (https://www.gutcellatlas.org/), Tabula Sapiens[42] (https://tabula-sapiens-portal.ds.czbiohub.org/) and Gene Expression Omnibus (GSE125970 (ref. [43]) and GSE116222 (ref. [44]), read counts and cluster results downloaded from the HPA: https://www.proteinatlas.org/about/download). For single-cell RNA-seq datasets, relative read counts were normalized

using Seurat package (v4.1.1)[73] in R, using 'Relative count' methods with a scale factor of $10^4$, and averaged across all cells. To compare the *APOBEC1* expression level in small and large intestine epithelial and stem cells, negative binomial regression models were constructed to see if difference exists after controlling confounding factors including number of mRNA counts in each cell, number of features in each cell and other APOBEC family gene expression. The code for this analysis can be found at: https://github.com/YichenWang1/small_bowel/tree/main/Expression.

### Reporting summary

Further information on research design is available in the Nature Portfolio Reporting Summary linked to this article.

## Data availability

DNA sequencing data generated for this study are deposited in the European Genome-Phenome Archive (EGA) with accession code EGAD00001008764. Existing DNA sequencing datasets used in the study are deposited in EGA with accession code EGAD00001004192 (PD37449, PD34200 and PD37266) and EGAD00001006641 (PD28690, PD43850 and PD43851). Existing RNA sequencing datasets were downloaded from Gut Cell Survey (https://www.gutcellatlas.org/), Tabula Sapiens (https://tabula-sapiens-portal.ds.czbiohub.org/) and Gene Expression Omnibus (GSE125970 and GSE116222, read counts and cluster results downloaded from the Human Protein Atlas: https://www.proteinatlas.org/about/download). The cBioPortal MutationMapper database used to annotate cancer hotspot mutations was accessed at https://www.cbioportal.org/mutation_mapper?standaloneMutationMapperGeneTab=ATM.

## Code availability

Code required to reproduce the analyses in this paper is available online. Mutation-calling algorithms are available through GitHub (https://github.com/cancerit). Variant calling filters can be found at https://github.com/MathijsSanders/SangerLCMFiltering and https://github.com/TimCoorens/Unmatched_NormSeq. All other custom code used in this study is available online at https://github.com/YichenWang1/small_bowel.

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

## Acknowledgements

We thank the staff of the Wellcome Sanger Institute Sample Logistics, Genotyping, Pulldown, Sequencing and Informatics facilities for their many contributions, especially L. O'Neill, Y. Hooks, C. Latimer, K. Roberts and I. Whitmore for their support with sample management and laboratory work.We thank S. Moody, E. Dunstone, R. Rahbari, S. Behjati and F. Markowetz (University of Cambridge and Cancer Research UK Cambridge Institute) for discussion of the results, C. Suo, W. Wang (Technical University of Munich), W. Zhao (Karolinska Institutet) for discussion regarding single-cell RNA-seq data analysis, S. Olafsson and M. Przybilla for discussion regarding statistical analyses. We gratefully acknowledge the contributions to this study made by the University of Birmingham's Human Biomaterials Resource Centre, which has been supported through Birmingham Science City—Experimental Medicine Network of Excellence Project; especially, we thank S. Beard and E. Hurlestone for coordination. We thank Cambridge University Hospitals NHS Foundation Trust Tissue Bank for assistance in the acquisition of samples, especially J. R. Davies for assistance in clinical data collection. We thank members of the Phoenix Consortium, in particular I. Debiram-Beecham and N. Grehan for their help with patient recruitment and sample collection. We also thank Royal Papworth Hospital NHS Trust Mortuary, in particular, M. Goddard and S. Preston for help with sample acquisition. We thank L. A. Aaltonen (University of Helsinki) for coordinating small bowel cancer data analysis. We thank all the patients and their families, without their support this work would not have been possible. This research is supported by core funding from the Wellcome Trust (206194). S.J.A.B is funded by a Cancer Research UK Advanced Clinician Scientist Fellowship (C14094/A27178) and The Pharsalia Trust. M.Z. is funded by a Medical Research Council (MRC) New Investigator Research Grant (MR/T001917/1) and NIHR Cambridge Biomedical Research Centre. L.M. is funded by a Jean Shank/Pathological Society of Great Britain and Ireland Intermediate Clinical Fellowship (JSPS IF 2019 01). I.M. is funded by Cancer Research UK (C57387/A21777) and the Wellcome Trust. The laboratory of R.C.F. is funded by a program grant from the Medical Research Council (RG84369). Y.W. and T.H.H.C. are supported by Wellcome Ph.D. Studentships and P.S.R. by a Wellcome Clinical Ph.D. fellowship. R.K. is supported by grants from Academy of Finland (Finnish Center of Excellence Program 2018–2025, No. 312041) and Cancer Society of Finland (200071). The GTEx Project is supported by the Common Fund of the Office of the Director of the National Institutes of Health, NCI, NHGRI, NHLBI, NIDA, NIMH and NINDS. The GTEx data used for the analyses described in this manuscript were obtained from the HPA Project. For the purpose of open access, the authors have applied a CC-BY public copyright license to any author-accepted manuscript version arising from this submission.

## Author contributions

Y.W., M.R.S., T.H.H.C. and P.S.R. conceived the study design. Y.W. and T.H.H.C. wrote the scripts. Y.W. performed the analyses with help and input from T.H.H.C., H.J. and R.K. Y.W., P.S.R., L.M., H.L. and A.N. undertook the laboratory work. H.L. and M.A.S. contributed to the analysis pipeline. P.S.R., S.J.A.B., M.Z., S.R., A.N., K.S., R.C.F., R.W., D.R., B.N., R.H. and R.B. recruited patients and provided small bowel biopsies and clinical information. M.R.S., P.J.C. and I.M. oversaw statistical analysis. M.R.S. oversaw the study. Y.W. and M.R.S. wrote the manuscript with input from all other authors.

## Competing interests

The authors declare no competing interests.

## Additional information

**Extended data** is available for this paper at https://doi.org/10.1038/s41588-022-01296-5.

**Correspondence and requests for materials** should be addressed to Michael R. Stratton.

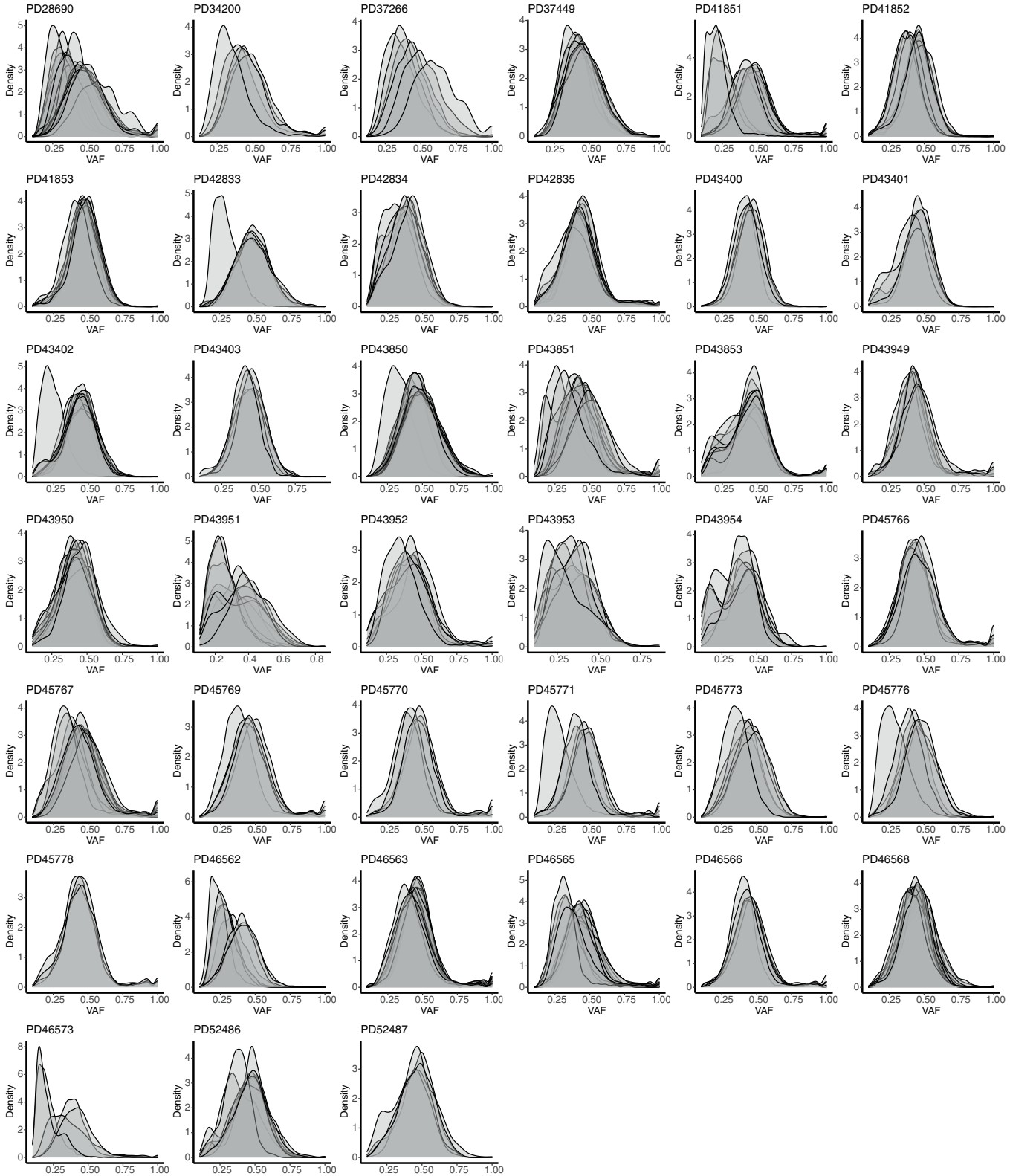

**Extended Data Fig. 1 | Variant allele frequency distributions reflecting clonality of LCM samples.** Variant allele frequency distribution for all individuals in this study. Most crypts have peaks around 0.5, indicating their monoclonity.

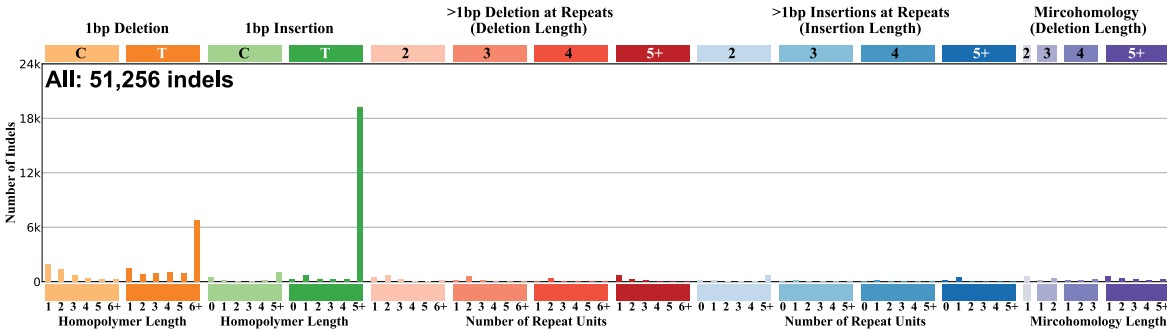

**Extended Data Fig. 2 | Indel spectrum for small intestinal crypts.** Number indicates total number of indels detected. 1 bp deletion and insertion at 6+ T homopolymers are the most common types of Indels.

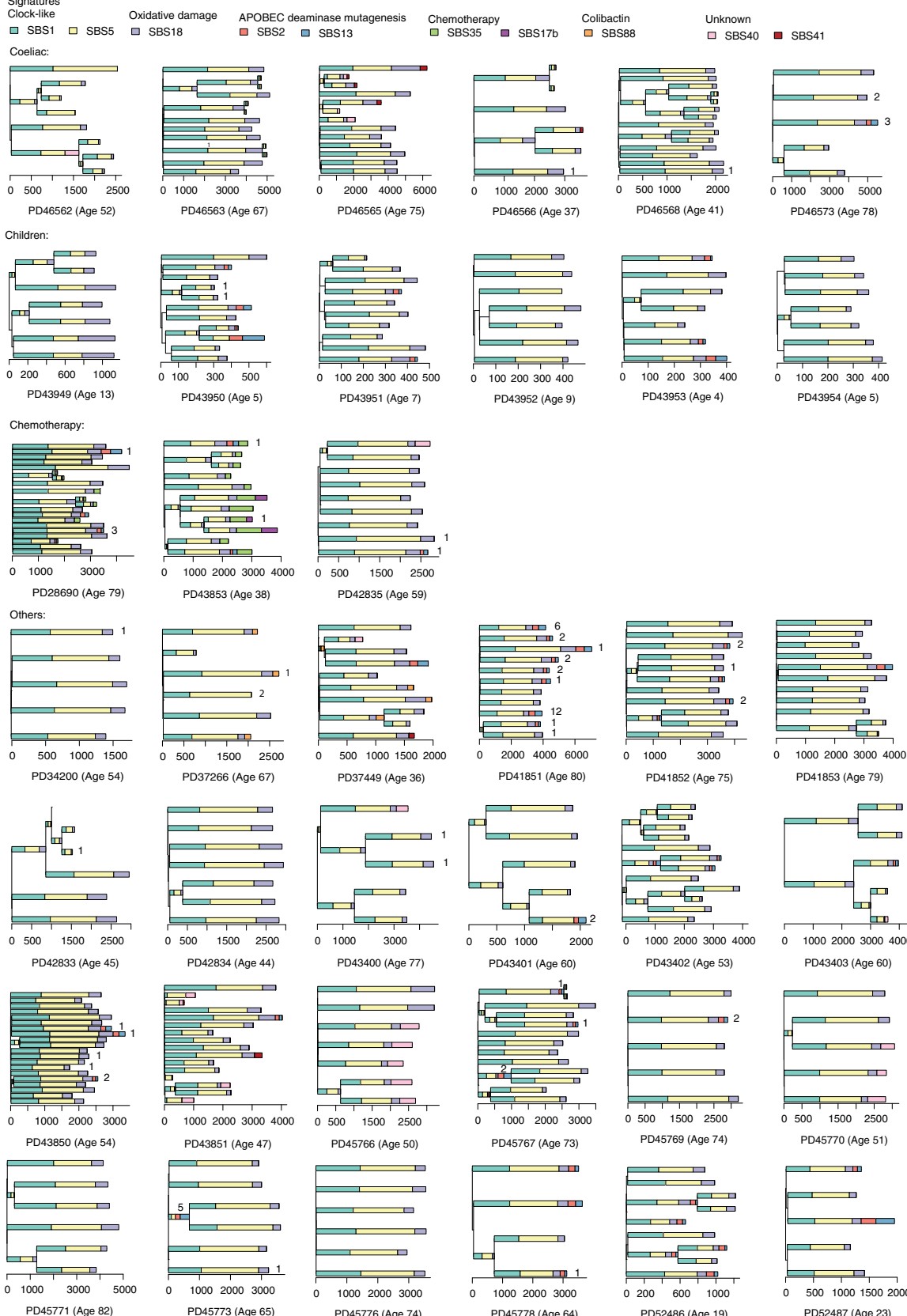

**Extended Data Fig. 3 | Phylogenies of all individuals in this study with mutational signature annotation.** Branch lengths correspond to SBS burdens, and color codes for mutational signatures are at the top. Numbers on the tips/branch indicate the number of hypermutation clusters placed on the tips/branch.

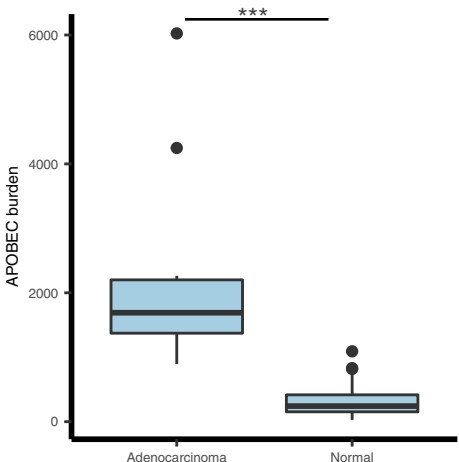

**Extended Data Fig. 4 | Comparison of APOBEC mutagenesis in small bowel cancer and normal crypts.** Boxplot of SBS2/13 exposures in small bowel adenocarcinomas and normal crypts with APOBEC mutagenesis (n = 14 for adenocarcinomas and n = 58 for normal crypts). The central line, box and whiskers represent the median, interquartile range (IQR) from first to third quartiles, and 1.5 × IQR, respectively. Burdens in cancer WES data have been adjusted by the proportion of exomes in genome to compare with whole-genome sequencing data. Median = 1691 (adenocarcinoma) and 242 (normal), two-tailed $t$-test $P = 2 \times 10^{-4}$.

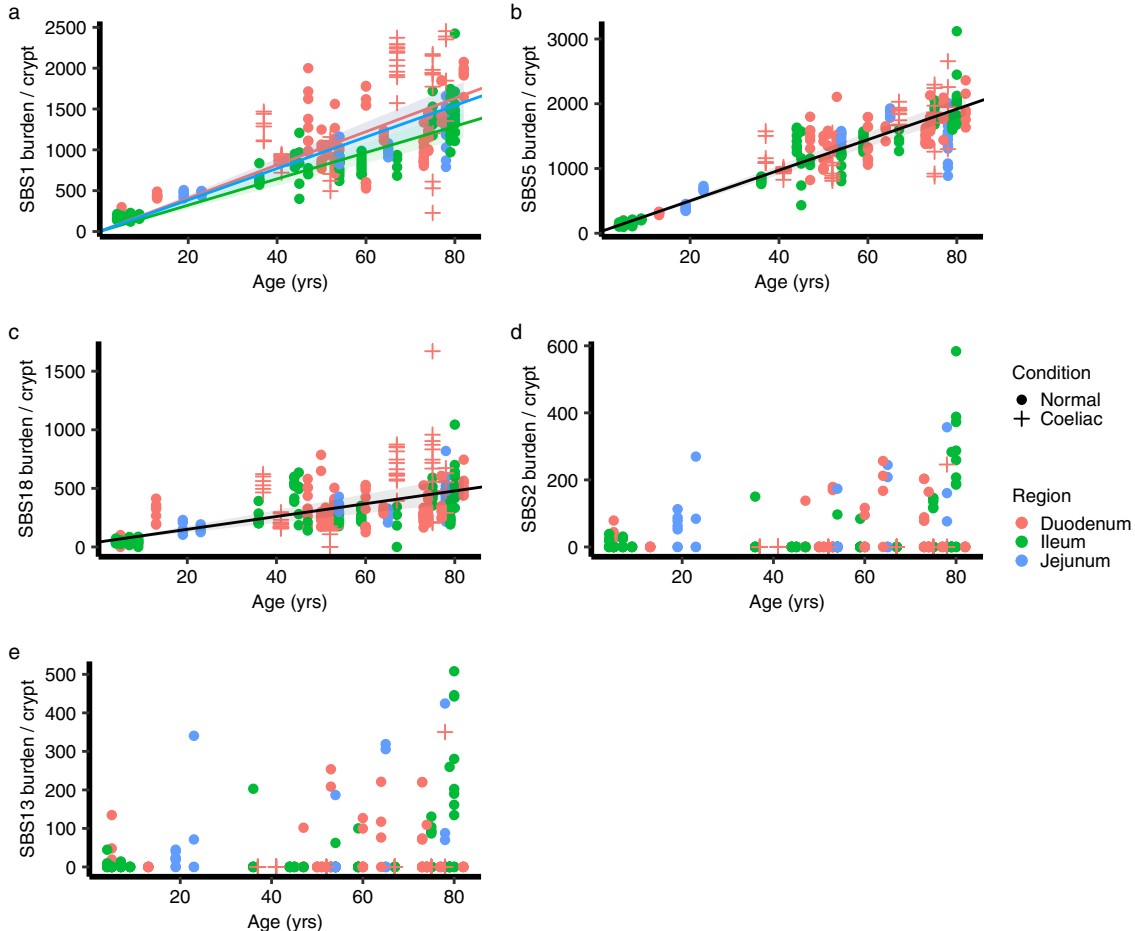

**Extended Data Fig. 5 | Signature burden versus age for SBS1, SBS5, SBS18, SBS2 and SBS13.** Regression lines were estimated using linear mixed models. Error bands represent 95% confidence interval for the fixed effect of age. Colors indicate biopsy regions, with orange, green and blue representing duodenum, ileum and jejunum, respectively. Shapes indicate whether the donor has a celiac history or not. Crosses indicates donors with a celiac history, and dots indicate donors without a celiac history. (**a**) SBS1 burden versus age, showing regression lines for the three different sectors of the small intestine. (**b**) SBS5 burden versus age, showing a regression line for all samples because the rate is not statistically different for the three sectors according to linear mixed models. (**c**) SBS18 burden versus age, showing a regression line for all samples because the rate is not statistically different for the three sectors according to linear mixed models. (**d**) SBS2 burden versus age, the relationship is not linear. (**e**) SBS13 burden versus age, the relationship is not linear.

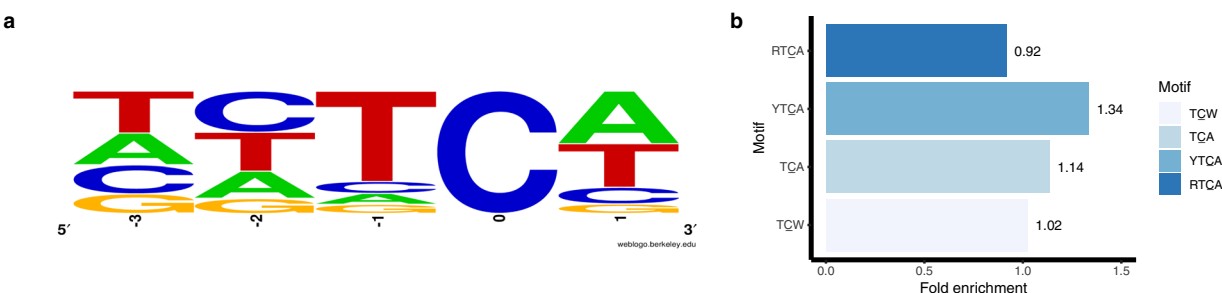

**Extended Data Fig. 6 | Extended contexts of APOBEC mutations.** Enrichment of pyrimidines (Y) instead of purines (R) at −2 position indicate APOBEC3B is unlikely to be the major contributing enzyme, while APOBEC1/3A could not be excluded. (**a**) Sequence logo showing the base frequency from −3 to +1 of APOBEC signature from HDP *de novo* extraction. (**b**) Fold enrichment of different TC contexts in all phylogenetic branches with SBS2/13, showing a preference for YTCA/TCA/TCW. RTCA: $P = 1$, YTCA: $P = 3.0 \times 10^{-96}$, TCA: $P = 2.3 \times 10^{-31}$, TCW: $P = 6.4 \times 10^{-4}$, one-tailed Fisher's exact test.

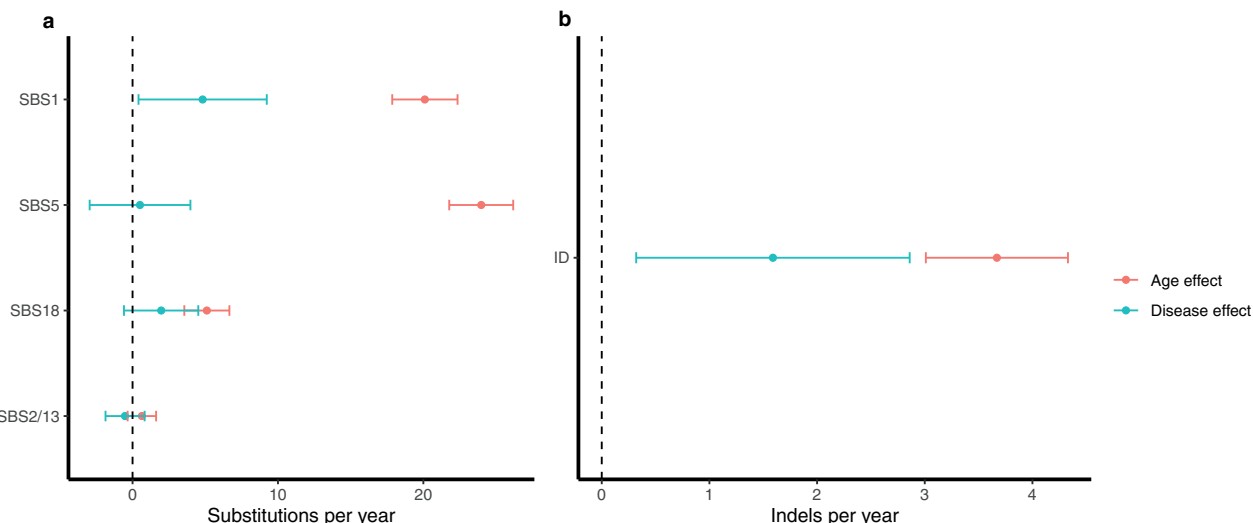

**Extended Data Fig. 7 | Comparisons of the effects of age and celiac disease on substitutions and Indels.** Central dots represent the estimated fixed effect from linear mixed-effects models, and error bars represent the 95% confidence intervals. N = 306 crypts. (**a**) Age and celiac disease effect on different single-base substitution mutational signatures. (**b**) Age and celiac disease effect on Indels.

| | |
|---|---|

# Reporting Summary

Please do not complete any field with "not applicable" or n/a. Refer to the help text for what text to use if an item is not relevant to your study.
For final submission: please carefully check your responses for accuracy; you will not be able to make changes later.

## Statistics

For all statistical analyses, confirm that the following items are present in the figure legend, table legend, main text, or Methods section.

| n/a | Confirmed | |
|---|---|---|
| ☐ | ☒ | The exact sample size (*n*) for each experimental group/condition, given as a discrete number and unit of measurement |
| | | A statement on whether measurements were taken from distinct samples or whether the same sample was measured repeatedly |
| ☐ | ☒ | The statistical test(s) used AND whether they are one- or two-sided |
| | | *Only common tests should be described solely by name; describe more complex techniques in the Methods section.* |
| ☐ | ☒ | A description of all covariates tested |
| ☐ | ☒ | A description of any assumptions or corrections, such as tests of normality and adjustment for multiple comparisons |
| ☐ | ☒ | A full description of the statistical parameters including central tendency (e.g. means) or other basic estimates (e.g. regression coefficient) AND variation (e.g. standard deviation) or associated estimates of uncertainty (e.g. confidence intervals) |
| ☐ | ☒ | For null hypothesis testing, the test statistic (e.g. *F*, *t*, *r*) with confidence intervals, effect sizes, degrees of freedom and *P* value noted *Give P values as exact values whenever suitable.* |
| ☐ | ☒ | For Bayesian analysis, information on the choice of priors and Markov chain Monte Carlo settings For hierarchical and complex designs, identification of the appropriate level for tests and full reporting of outcomes |
| ☐ | ☒ | Estimates of effect sizes (e.g. Cohen's *d*, Pearson's *r*), indicating how they were calculated |
| ☒ | ☒ | *Our web collection on statistics for biologists contains articles on many of the points above.* |

## Software and code

Policy information about availability of computer code

| Data collection | No code and software was required to collect data. |
|---|---|
| Data analysis | Mutation-calling algorithms are available through GitHub (https://github.com/cancerit). Variant calling filters can be found at https://github.com/MathijsSanders/SangerLCMFiltering and https://github.com/TimCoorens/Unmatched_NormSeq .<br><br>Statistical analysis was performed in R (3.0.4 and 4.0.2). All custom code for this study is available on GitHub at https://github.com/YichenWang1/small_bowel. Any additional code will be provided by the authors on request.<br><br>Open source R packages:<br>HDP - version 0.1.5 (https://github.com/nicolaroberts/hdp)<br>sigFit - version 2.0 (https://github.com/kgori/sigfit)<br>nlme - version 3.1-148 CRAN<br>ggtree - version 3.3.1 Bioconductor<br>Seurat - version 4.1.1 CRAN<br>ape - version 5.6.1 CRAN<br>GenomicRanges - version 1.42.0 Bioconductor<br>BSgenome - version 1.58.0 Bioconductor<br>BSgenome.Hsapiens.UCSC.hg19 - version 1.4.3 Bioconductor<br>dplyr - version 1.0.7 CRAN<br>tidyr - version 1.2.0 CRAN<br>tidyverse - version 1.3.1 CRAN<br>RColorBrewer - version 1.1.2 CRAN<br>ggplot2 - version 3.3.5 CRAN |

knitr - version 1.37 CRAN

Additional publicly available code and software used:
BWA-Mem - version 0.7.12-r1039 and 0.7.17 (https://github.com/lh3/bwa)
CaVEMan - versions 1.11.2, 1.14.1, 1.15.1(https://github.com/cancerit)
cgpPindel - versions 2.2.2, 3.3.0 (https://github.com/cancerit)
ASCAT - versions 4.0.1, 4.1.2, 4.5.0 (https://github.com/cancerit)
Battenberg - versions 3.5.3 (https://github.com/cancerit)    BRASS
- version 6.1.2, 6.3.4 (https://github.com/cancerit)
GRIDSS - version 2.13.1 (https://github.com/PapenfussLab/gridss)
JBrowse - version 1.15.2 (https://jbrowse.org/blog/2018/08/16/jbrowse-1.15.2-maintenance-release-index.html)
MPBoot - version 1.1.0 (https://github.com/diepthihoang/mpboot)

For manuscripts utilizing custom algorithms or software that are central to the research but not yet described in published literature, software must be made available to editors and reviewers. We strongly encourage code deposition in a community repository (e.g. GitHub). See the Nature Portfolio guidelines for submitting code & software for further information.

# Data

Policy information about availability of data

All manuscripts must include a data availability statement. This statement should provide the following information, where applicable:
- Accession codes, unique identifiers, or web links for publicly available datasets
- A description of any restrictions on data availability
- For clinical datasets or third party data, please ensure that the statement adheres to our policy

DNA sequencing data generated for this study are deposited in the European Genome-Phenome Archive (EGA) with accession code EGAD00001008764. There is no restriction on data availability. Existing DNA sequencing data used in study are deposited in EGA with accession code EGAD00001004192 (PD37449, PD34200, PD37266) and EGAD00001006641 (PD28690, PD43850, PD43851). Existing RNA sequencing data were downloaded from Gut Cell Survey (https://www.gutcellatlas.org/), Tabula Sapiens (https://tabula-sapiens-portal.ds.czbiohub.org/), and Gene Expression Omnibus (GSE125970 and GSE116222, read counts    and cluster results downloaded from the Human Protein Atlas: https://www.proteinatlas.org/about/download).
The cBioPortal MutationMapper database used to annotate cancer hotspot mutations was accessed at:
https://www.cbioportal.org/mutation_mapper?standaloneMutationMapperGeneTab=ATM.

# Field-specific reporting

Please select the one below that is the best fit for your research. If you are not sure, read the appropriate sections before making your selection.

☒ Life sciences   ☐ Behavioural & social sciences   ☐ Ecological, evolutionary & environmental sciences

For a reference copy of the document with all sections, see nature.com/documents/nr-reporting-summary-flat.pdf

# Life sciences study design

All studies must disclose on these points even when the disclosure is negative.

| | |
|---|---|
| Sample size | No sample-size calculation was performed. Sample size was determined by the availability of tissue and cost of the experiment. |
| Data exclusions | One crypt (PD42835_lo0032) was excluded from the start due to clear evidence of contamination. All other samples were included in downstream analysis and had their mutational landscape reported. When modelling mutation rates, only samples with >15-fold coverage were included, and one patient (PD43853) with a substantial number of mutations due to chemotherapy was excluded. Two biopsies (PD43851j_P52_DDM_E2, PD46565c_lo0009) have a mutational landscape dominant by SBS5/40 and with lower mutation rates, which is distinct from the mutational landscape of normal small intestinal crypts, and similar to that of Brunner's glands, despite their crypt-like appearance under microscopy inspection. These two samples was kept and reported for the comprehensiveness and transparency of this dataset, but were not included in the statistical modeling, and we ran a separate signature extraction on the remaining crypts, to avoid interference from the two unusual cases. When modelling clonal dynamics, crypts from coeliac patients, children and the patient with excessive amount of mutations from chemotherapy (PD43853) were excluded, because these crypts might have different number of cells and non-constant mutation rate. |
| Replication | Our manuscript describes the mutational landscape of the normal tissues, it does not test specific hypotheses, and so replication does not apply in its usual way. Sequencing replicates are not normally possible as most of these samples have been obtained from distinct crypts, though mutation rates and signatures for crypts from the same individual demonstrated good concordance. Validation of the low-input whole genome sequencing protocol is detailed in Ellis et al 2021, and the protocol was applied in previously published work. Validation of low-input whole genome sequencing after crypt isolation is available as part of the studies published in Lee-Six et al 2019 and Moore et al 2020. Ellis et    al 2021. |
| Randomization | Not applicable - cases were not subjected to any intervention. Covariates such as age, biopsy location and disease condition were controlled in statistical modelling. |
| Blinding | Not applicable - cases were not subjected to any intervention and were not allocated into groups. For pre-existing conditions, it was not |

| Blinding | possible for the researchers to remain fully unaware of the conditions, because they were self-explanatory during the collection and analysis: for the coeliac samples, pathological morphology such as disruption of villi structures were observed during laser capture microdissection; for patients undergone chemotherapy, this was revealed by chemo-therapy related mutational signatures left on their genomes. |
|---|---|

# Reporting for specific materials, systems and methods

We require information from authors about some types of materials, experimental systems and methods used in many studies. Here, indicate whether each material, system or method listed is relevant to your study. If you are not sure if a list item applies to your research, read the appropriate section before selecting a response.

## Materials & experimental systems

| n/a | Involved in the study |
|---|---|
| ☒ ☐ | Antibodies |
| ☒ ☐ | Eukaryotic cell lines |
| ☒ ☐ | Palaeontology and archaeology |
| ☒ ☐ | Animals and other organisms |
| ☒ ☐ | Human research participants |
| ☐ ☒ | Clinical data |
| ☒ ☐ | Dual use research of concern |
| ☒ ☐ | |

## Methods

| n/a | Involved in the study |
|---|---|
| ☒ ☐ | ChIP-seq |
| ☒ ☐ | Flow cytometry |
| ☒ ☐ | MRI-based neuroimaging |

# Human research participants

Policy information about studies involving human research participants

| Population characteristics | The dataset includes 342 individual small intestinal crypts together with three Brunner's glands from 39 individuals (22 females and 17 males aged between 4 - 82 years from UK). Among them six individuals (two females, four males) had a history of coeliac disease (gluten enteropathy). Information for each individual is provided in Supplementary table 6. |
|---|---|
| Recruitment | Participants were recruited from eight cohorts, including organ donors (providing duodenum, jejunum and ileum samples), patients who have undergone endoscopy (providing duodenum samples) and colorectal cancer patients who have had surgical removal for part of their colon (including part of the ileum). Samples were selected to cover a wide age range and no other selection criteria were applied. In retrospect, we have a good balance of gender (22 females, 17 males). However, all participants were based in UK, and this might need to be considered when generalise the conclusions to other regions and ethic groups. To be specific, our first cohort consists of samples from a 78-year-old man (PD28690), a 54-year-old woman (PD43850) and a 47-year-old man (PD43851) in warm autopsies within 6h of death (REC 13/EE/0043 and 17/LO/1801). Samples in the second cohort were from organ donors (two females, one male) at age 36 to 67 from whom small-intestinal biopsies were taken at the time of organ donation (REC 15/EE/0152). The third cohort represents four female and three male patients aged 38 to 80 who underwent surgical resection (REC 15/WA/0131). The fourth represents four female patients aged 53 to 77 who had endoscopy (REC 08/h0304/85+5). The fifth cohort are children between 4 to 13 (four females, two males) who had endoscopy (REC 17/EE/0265). The sixth cohort comprised six male and two female patients aged 50 to 82 with surgical resection (REC 20/NW/0001). The seventh cohort are two female and four male patients aged from 37 to 78 with coeliac conditions, and samples were collected through endoscopy (REC 18/ES/0133). The last cohort are from AMSBio (commercial supplier), samples for donors PD52486, PD52487 were obtained at autopsy from two female individuals who had died of causes not related to cancer (REC 17/LO/1801). |
| Ethics oversight | National Research Ethics Service Committee East of England: PD28690, PD37449, PD34200, PD37266<br>London-Surrey Research Ethics Committee: PD43850, PD43851, PD52486, PD52487<br>Wales REC 7: PD41851, PD41852, PD41853, PD42833, PD42834, PD42835, PD43853<br>NRES Committee East of England – Cambridge East: PD43400, PD43401, PD43402, PD43403<br>NRES Committee East of England – Cambridge South: PD43949, PD43950, PD43951, PD43952, PD43953, PD43954<br>NRES Committee North West – Haydock: PD45766, PD45767, PD45769, PD45770, PD45771, PD45773, PD45776, PD45778<br>East of Scotland Research Ethics Service : PD46562, PD46563, PD46565, PD46566, PD46568, PD46573 |

Note that full information on the approval of the study protocol must also be provided in the manuscript.

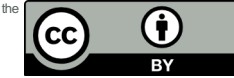

