## [Peer Review File · Nature Genetics]

Peer Review Information

Manuscript Title: APOBEC mutagenesis is a common process in normal human small intestine

Corresponding author name(s): Professor Michael Stratton

Reviewer Comments & Decisions:

Decision Letter, initial version:
--

6th Jun 2022

Dear Professor Stratton,

First of all, please accept my apologies for the delay in returning this decision to you. Thank you for bearing with me.

Your Letter entitled "APOBEC mutagenesis is a common process in normal human small intestine" has now been seen by 3 referees, whose comments are attached. While they find your work of potential interest, they have raised concerns which in our view are sufficiently important that they preclude publication of the work in Nature Genetics, at least in its present form.

Overall, the reviewers all agree that this is a well-written and technically robust paper. However, Reviewers #1 and #2 are both of the view that the biological novelty afforded by the paper is limited. We were encouraged by the enthusiasm shown by Reviewer #3 (who has only made requests for more experimental details) but having considered the three reports holistically, also taking into account reviewer expertise, we believe that the paper has not received the level of support we'd expect at this stage. As such, we won't be able to pursue it in its present form, but we'd be very happy to consider a revised version that provides a more mechanistic exploration of some of your intriguing findings. Reviewer #2 has made some suggestions about potential avenues to explore, but please don't feel beholden to this list. If there are other aspects that you wish to focus on, please feel free to do so – at a high level, the reviewers are asking for a more developed biological narrative, and I think that there are likely several routes to achieve this goal.

I do hope that you'll pick this option and come back to us with an appeal – I'd very much like to see the manuscript again, although please understand that until we have read the revised manuscript in its entirety we cannot promise that it will be sent back for peer review. If you are interested in revising this manuscript for submission to Nature Genetics in the future, please contact me to discuss a potential appeal.

Alternatively, if you are in a hurry to publish, then I'd be happy to consult with my colleagues at Nature Communications to see if they'd be interested in the paper. This would involve sharing the paper, the review reports, and the reviewer identities which would allow the editorial team to decide whether they'd like to pursue the paper, and also which revisions they'd require you to do. If this would be helpful to you, please let me know and I'll get the ball rolling.

I look forward to hearing from you.

Sincerely,

Safia Danovi
Editor
Nature Genetics

Referee expertise:

Referee #1: somatic mutagenesis and evolution

Referee #2: intestinal carcinogenesis

Referee #3: APOBEC (please note that this reviewer has provided an attachment)

Reviewers' Comments:

Reviewer #1:

Remarks to the Author:

In the manuscript "APOBEC mutagenesis is a common process in normal human small intestine", Wang et al. perform whole genome sequencing of 342 human small intestinal crypts. They find that the mutation rate does not much differ from the rate observed in the colon, excluding differential mutation rates as a potential reason for the diverging cancer incidence in the small vs. the large intestine. Mutational signatures in general also behave similarly as in the colon, except there is a larger frequency of crypts that have experienced APOBEC mutagenesis (SBS2/SBS13) in the small intestine.

This paper is very well written and exceedingly logically structured, making it a joy to read. The underlying data set of 342 crypt genomes is also impressive. However, I regret to say that I was not able to identify any substantial novel results in the manuscript. The authors apply what are now standards methods (most of them developed by their own group): crypt microdissection and sequencing, calling of somatic variants, mutational signature extraction, mutation rate calculation, phylogenetic reconstruction, and come up with a small portfolio of observations. Methodologically, there is little to nothing to criticize – there is no doubt that the methods are robust, and the research is very well executed. However, the findings are very limited. Specifically,

1) It is interesting to read that the mutation rate in the small intestine does not really deviate from

the colon. The authors might have hypothesized at the beginning of the study that they would find lower mutation rates in the small intestine, perhaps partially explaining the reduced cancer risk. However, this is not the case. I think this result is important, but it is essentially negative and also not entirely novel as the authors have already published this result in one of their prior papers (Moore et al., Nature 2021), albeit backed by a smaller data set.

2) The patchy nature of APOBEC mutagenesis has also been previously published by the same group (Lawson et al., Science 2020). I acknowledge it is a different organ now, but conceptually there appears to be no difference, and the authors do not appear to be any closer to being able to explain the mechanisms underlying these observations.

3) The results on kataegis in Figure 4 are also not very surprising given prior knowledge.

Other than these (fairly severe) concerns about novelty, I find the paper very polished and have no technical concerns. One minor thought was that it would be helpful if the authors could clarify the anatomical relationship between dissected crypts from the beginning (were they all adjacent and from the same tissue section or were some of them spatially separated?). Figure 3 shows adjacent crypts, but I was not sure whether that applied to all the microdissected specimens.

Reviewer #2:

Remarks to the Author:

This paper reproduces the considerable technological achievement of Lee Six et al, 2019 by undertaking 25X WGS on laser captured normal appearing crypts from the gut. This time the small intestine is assessed and undergoes a very similar analysis pipeline to that reported in the colon, including construction of phylogenetic trees, comparison with tissue specific cancers and analysis of mutation signatures (indicating a higher SB frequency of characteristic APOBEC mutations).

Although the technical achievement of producing WGS data from such a small amount of human tissue should not be underestimated, the reproduction of the technique and the analysis pipeline from the previous publication inevitably impacts the novelty of the work. Some interesting observations are made, but these are not explored mechanistically or at depth and the paper, somewhat frustratingly, leaves the reader with more questions than it answers.

Specific comments.

1. Mutations characteristic for APOBEC are seen at higher frequency in the small intestine than in the colon but this interesting phenomenon is not investigated in any depth. APOBEC1 has tissue restricted expression in the small intestine (Hirano et al, 1997). Is there any similar differential expression of APOBEC3A expression along the length of the human gut?

2. Can mutation burden/phylogenetic trees be used as substrate for models to infer human small intestinal stem cell dynamics or clonal expansion and contrast with that in the colonic data of Lee Six et al? (as per work of Doug Winton, Ben Simons)

3. Following on from this - Mutation rates are found to be comparable to human colon, but was this normalised for putative differences in colonic and small intestinal crypt stem and progenitor cell number?.

4. A post chemo patient had a much higher mutation rate and was excluded from analysis, which feels like an opportunity lost. Why not further explore this interesting observation to draw inferences about cell dynamics/turnover in the gut following cytotoxic drug therapy or following radiation enteritis.

5. It is mentioned that coeliac didn't impact mutation burden in phenotypically normal crypts, but we were told very little about the disease state. Was there characteristic crypt hyperplasia seen in these patients or had this resolved post treatment?.

6. Is stable mutation rate and lack of clonal expansion also the case in other small intestinal conditions such as Crohns disease?.

7. The difference in cancer prevalence between the small bowel and colon is mentioned in passing but no real attempt is made to use these data and that available from Lee Six et al, 2019 to explore this longstanding and very important question.

In summary, the paper replicates the considerable technological achievement of previous work in a different organ, but fails to truly exploit the capacity to assess normal crypt mutation burden to further or mechanistically explore some of the interesting observations that are made.

Reviewer #3:

Remarks to the Author:

Manuscript describes a study of whole-genome spectra of changes in over 300 individual small intestinal crypts obtained by microdissection from tissue biopsies of 39 individuals. Each microdissection collected clonal group of cells which allows sufficient accuracy of calling somatic genome changes present in a cell that gave rise to a crypt-clone. Most frequent changes were single base substitutions (SBS). Agnostic signature extraction as well as comparison with previously identified SBSes revealed several mutational signatures of unknown origin, which could be caused by either endogenous or by exogenous DNA lesions. They also found two well defined endogenous mutational processes, clock-like SBS1 stemming from deamination of 5me-C in CpG motifs and a signature of cytidine deamination by single-stranded DNA-specific APOBEC cytidine deaminases. The latter signature is one of the most abundant in human cancers, however, was rarely found in normal somatic cells (so far on bladder and bronchial epithelia). Analysis of extended APOBEC mutational context indicated that the APOBEC3A is the likely source of mutations. The presence of APOBEC mutagenesis was supported by observation of mutation clusters highly enriched with APOBEC mutational motifs. Detailed analysis of adjacent crypts established that APOBEC mutagenesis is not uniformly distributed across small intestinal epithelium but is rather episodic. The conclusions about the presence and the features of APOBEC mutagenesis are of interest to the fields of genome stability, mutagenesis, and cancer biology.

Note:

Since pages in the Manuscript file were not numbered positions in the manuscripts are referenced by quotations.

Critique.

1. Insufficient source data and description of analysis do not allow recapitulation and reuse of results, and validation of the conclusions.

A. Insufficient source data on mutation call statistics and signatures in individual samples.

In this study, multiple samples (crypts) were sequenced from each individual. Supplementary Table 4 lists metadata for individuals, but not for sequenced crypts. However, all display items supporting conclusions of the manuscript contain information derived from sequencing genomes of individual samples/crypts. The Supplementary Table listing all sequenced crypts, their anatomical origin, numbers of SBS, SV, ID, CNV for each sequenced sample as well as SBS signature mutation load determined for each crypt is required. It took a while from me to locate such a Table on https://github.com/YichenWang1/small_bowel/blob/main/data/stat_summary.txt, and I do not see any reason for making any interested reader to go through this search instead of just placing this source data for easy access into Supplementary part. I attached Excel version of this Table "Crypts_stat_summary.txt.xlsx" for convenience of all parties involved in reviewing or/and handling this manuscript. Authors should also provide Readme explaining the content of each column and refer in Figure legends and/or in Methods to the columns containing values used for creating each display item. So far there are some inconsistencies between this Table and Figures or/and Method description. The inconsistencies found by this reviewer are in Specific Points section.

B. Lack of mutation call information used as direct input into signature and mutation cluster analyses.

Github folder https://github.com/YichenWang1/small_bowel/tree/main/data/vcf contains mutation calls for each sequenced sample/crypt. However, in Methods they state:

"Before running HDP [signature extraction pipeline], we assigned mutations to branches on the phylogenetic tree. This avoided counting the same mutation within one patient multiple times. To avoid overfitting, we only kept branches with > 50 mutations as input."

This implies that mutations in the trunk of a phylogenetic tree that, based on Extended Data Figure 4, were present in up to four individual samples were assigned to a single sample from a group joined by common phylogeny. Sample specific mutation catalogues in VCF or MAF format adjusted by removal of duplicated calls should be provided as a source data to allow recapitulation and reuse of results, and validation of the conclusions. While authors provide scripts that were used for analysis, it would require unreasonable effort from interested colleagues to recreate deduplicated VCF or MAF catalogues. Moreover, it would not be possible to generate the same deduplicated catalogues, because assignment of the trunk to a single catalogue was made by authors based on an undisclosed rule or just randomly.

2. Arbitrary threshold of 5% for APOBEC SBS2/13 as well as other SBS signatures for designating a sample as a "signature positive" (Methods sections "Mutational signature extraction..." and "Comparison with the small bowel adenocarcinoma...")

Note that percentage would depend not only on the number of APOBEC motif mutations in C (G) but also on other C (G)-motifs as well as on the number of T (A). Authors may want to try enrichment statistically evaluated for the excess of APOBEC motif C (G) mutations over the presence of APOBEC

motifs in surrounding genomic background. For the enrichment with APOBEC mutational motif a rigorous statistical evaluation does exist (refs 27, 44, 45 and <https://github.com/NIEHS/P-MACD>). This method can also detect APOBEC mutational signature in clusters where it is more pronounced than in whole-genome catalogue.

3. Cluster analysis

A. Information about the content of deduplicated sample-specific mutation catalogues is essential for mutation cluster (kataegis) calling. The Methods section "Identifying kataegis" does not describe how authors avoided using several times the calls belonging to the trunk of a phylogeny tree. This should be clarified in methods with reference to deduplicated mutation catalogues.

B. What was the minimal number of mutations in a group with less than 10 kb inter-mutational distance that was evaluated as a potential cluster? References 27, 44, 45 as well as PMID 31568516 evaluated groups of at least 2 mutations. Strand-coordinated clusters with as little as 2 mutations showed enrichment with APOBEC mutational motif greater than scattered mutations in the same genomes.

C. There are several samples with APOBEC mutation clusters, but still APOBEC-negative genome-wide. Presence of APOBEC mutation cluster(s) is the evidence of APOBEC mutagenesis operating in the history of a sequenced genome. The lack of APOBEC mutagenesis detectable on a genome-wide scale may be due to low abundance of single-stranded DNA accessible to APOBEC enzymes. Presence of APOBEC clusters in the genome of each sample should be indicated on the Supplementary Table exemplified by "Crypts_stat_summary.txt.xlsx" and on illustrations of phylogeny trees. It is important to know if APOBEC clusters would be detected in the trunks and/or in the branches which are qualified as APOBEC negative.

Specific points.

1. "When modelling mutation rates, only samples with >15-fold coverage were included, and one patient (PD43853) with a substantial number of mutations due to chemotherapy was excluded (n = 306)"

Blacklisted samples should be specified in the Supplementary Table exemplified by "Crypts_stat_summary.txt.xlsx". So far I was unable to get 306 whitelisted samples applying filters quoted above.

2. "commonly found in many cancer types... and uterus (refs1,2)". Authors may want to add another reference #45 (2013) here, which used an alternative to NMF motif-specific approach to demonstrate ubiquitous presence of APOBEC mutagenesis in a number of human cancer types.

3. "However, the median burden of APOBEC mutations in normal small intestine was ~7-fold lower than that observed in small bowel adenocarcinoma"

Replace "the median burden of APOBEC mutations " with "SBS2/13 burden" to be consistent with Extended Data Fig. 6 legend and to clarify that was it the contribution of SBS2+SBS13 obtained as an

outcome of HDP, not the simple counts of APOBEC motif mutations. This is important to allow analyses in follow-up studies to be set consistent with this work.

4. "Enrichment of SBS2/13 mutations with a pyrimidine rather than a purine (ref 27)..."

Note that only peaks with A in -2 positions are labelled on Extended Data Fig 4; Peaks with another purine – G are not labeled, which makes it confusing to a reader

What values are on y-axis of Extended Data Fig 4? Is the statement of higher presence of pyrimidines over purines supported by statistics? Compare with the estimates obtained with the algorithm described in ref 27 and available at <https://github.com/NIEHS/P-MACD> . This algorithm generates p-values.

What is the preference in -2 position for mutations in APOBEC mutation clusters? Based on Ref 27 APOBEC3B may operate in all genomes, while more potent APOBEC3A mutagenesis may mask the impact of APOBEC3B. Note that APOBEC3B mutagenesis is readily detectable in mutation clusters of cancer genomes with low level of APOBEC mutagenesis (PMID 30537516).

5. "SBS2/13 mutations in cancer are found in localised, high-density clusters referred to as 'kataegis' (refs 29,30)"

Reference 44 - Roberts 2012 - also demonstrated strand coordinated-clusters with clear APOBEC mutational motif and was published simultaneously with reference 29.

6. Replace ref 18 from preprint to published paper.

7. "For identified SBS signatures, signatures with ≥ 0.9 cosine similarity with the reference were considered as the same signatures"

SBS2+SBS13 should be used for cosine similarity with extracted HDP signatures here and in the analyses shown in Extended Data Fig. 8j. In this figure, Component 4, which is clearly APOBEC signature has cosine similarities approx. 0.4 with either SBS2 or SBS13, however it is clear that the total of SBS2+SBS13 will show very high cosine similarity with Component 4.

8. Extended Data Fig. 4.

Color codes unreadable or/and indistinguishable. Increase size (possibly by making two panels) and put the SBS name on top of each section of the corresponding branch. Phylogeny trees support important conclusions of this manuscript, therefore source data with numeric values used to build this figure should be provided.

If you do decide to transfer to Nature Communications, please use our [redacted]

You will not have to re-supply manuscript metadata and files, unless you wish to make modifications, but please note that this link can only be used once and remains active until used. For more information, please see our http://www.nature.com/authors/author_resources/transfer_manuscripts.html?WT.mc_id=EMI_NPG_1511_AUTHORTRANSF&WT.ec_id=AUTHOR manuscript transfer FAQ page.

Note that any decision to opt in to In Review at the original journal is not sent to the receiving journal on transfer. You can opt in to [In Review](https://www.nature.com/nature-portfolio/for-authors/in-review) at receiving journals that support this service by choosing to modify your manuscript on transfer. In Review is available for primary research manuscript types only.

Author Rebuttal, Appeal

We sincerely thank the Referees for their comments and the substantial time and effort on our behalf they have expended in providing them.

General point

A question raised by two Referees was the degree of novelty and additional insight brought by our studies. The main finding in this study is the surprisingly high level of APOBEC DNA editing in the normal small intestine compared to normal large intestine and most other normal tissues. We explored this in multiple ways providing information on its sporadic nature, episodic pattern, age of onset, presence of kataegis, likely intrinsic origin and mutation burden compared to cancers. In our opinion, this is the most comprehensive and detailed analysis of APOBEC mutagenesis in normal cells in the literature.

In the previous submission we were unable, however, to provide an explanation for the very high levels of APOBEC mutagenesis in the small intestine epithelium compared to elsewhere. Subsequently, on review of wider relevant data we have, in our view, generated a highly plausible hypothesis for the observation. In general in human cells, SBS2 and SBS13 are believed to be generated by APOBEC3A, and to a lesser extent by APOBEC3B (see recent publications referred to in our paper). The small intestine epithelium is, however, unique among cell types in having very high levels of APOBEC1 expression (much higher than the colon and other tissues). The reason for this is thought to be the need for RNA editing by APOBEC1 of the APOB mRNA in order to generate a truncated APOB protein which is used in lipid trafficking. Indeed, this is where the name for the APOBEC enzyme family originates. As outlined in the new version of the manuscript, the sequence context of APOBEC1 induced mutations is in agreement with that found in the small intestine (and is similar to the APOBEC3A sequence context).

We therefore now believe that this is the reason for the surprising amount of SBS2/SBS13 in small intestine. This would indicate that APOBEC1 (in addition to APOBEC3A/B) can perform DNA editing in humans, that APOBEC1 is conducting both DNA and RNA editing in small intestine cells and potentially that the DNA editing is simply collateral damage consequent on the high levels of APOBEC1 expression required for its core RNA editing function.

Given the general current topicality of APOBEC mediated DNA editing and the limited knowledge of its role in normal cells, we hope that, added to the previously included data, this extra key insight provides the higher and broader level of interest that would justify publication in Nature Genetics.

We have added a main Fig. 5, extended Fig. 6 and Table 1 summarizing the level of APOBECs in small and large intestine across multiple datasets, as well as paragraphs in the main text discussing this in detail:

Line 223-251:

“Multiple lines of evidence, including the sequence context of SBS2/13 mutations in human cancers³⁵, engineered expression of APOBEC enzymes in model systems^{35,36}, and APOBEC gene knockouts³⁷ in human cancer cell lines, indicate that among the family of 11 APOBEC enzymes, APOBEC3A and, to a lesser extent, APOBEC3B are responsible for generating SBS2/13. To investigate the underlying mechanism for the unusually high level of SBS2/13 in small intestine epithelium and its sharp reduction on transition to the large intestine, we analyzed five independent datasets of publicly accessible bulk tissue³⁸ and single-cell transcriptome sequences^{39,40,41,42,43,44}. This revealed that a) *APOBEC1* is expressed at much higher levels in small intestine epithelium than large intestine epithelium (10-20 fold higher normalized TPM in bulk RNA sequencing data and ~10 fold higher relative read counts in single-cell RNA sequencing data of epithelium) and all other normal tissues (**Fig. 5** and **Extended Data Fig. 6**), and b) this difference in *APOBEC1* expression is even more pronounced between small and large intestine epithelial stem cells/transit amplifying cells in which the SBS2/13 mutations found here must have been generated (20-40 fold higher relative read counts). Moreover, four of the five datasets indicate that c) *APOBEC1* is expressed at higher levels than *APOBEC3A* and *APOBEC3B* in small intestine epithelium and d) in contrast to *APOBEC1*, that *APOBEC3A* and *APOBEC3B* show lower expression in small intestine than in large intestine epithelium (**Table 1**).

APOBEC1 has a known physiological function, mediating C>U editing at position 6666 in Apolipoprotein B (*APOB*) mRNA to generate a truncated APOB protein with specific lipid trafficking capabilities critical to normal lipid absorption and distribution by the small intestine^{45,46}. In addition to its RNA editing capability, in experimental systems *APOBEC1* generates C>T mutations in DNA with a sequence context similar to that of *APOBEC3A*^{47,48,49,50}, and consistent with that observed here in small intestine SBS2/13 mutations (**Extended Data Fig. 7**). Thus, it is plausible that the 10-40 fold higher level of *APOBEC1* mRNA expression in small compared to large intestine epithelium is responsible for the ~28-fold higher SBS2/13

frequency in small compared to large intestine epithelium, and the even greater differences compared to most other normal tissues. “

In the Discussion we have also added line 280-295:

“APOBEC1 has rarely been considered^{51,52} as a contributor to SBS2/13 mutation burden in cancer or normal tissues because of its small intestine specific expression profile. However, the association between the 10-40 fold differences in *APOBEC1* mRNA expression levels and the ~28-fold difference in SBS2/13 frequency comparing small and large intestine epithelia provides strong circumstantial evidence that APOBEC1 is responsible for the high SBS2/13 mutation levels in normal small intestine. Definitive examination of this hypothesis would be provided by *APOBEC1* knockout in organoids derived from normal small intestine epithelium, although if SBS2/13 mutation episodes are as infrequent *in vitro* as *in vivo*, these might be daunting experiments to conduct. If correct, however, this indicates that APOBEC1, in addition to APOBEC3A and APOBEC3B, can contribute to SBS2/13 mutations in human cells, and, therefore, that APOBEC1 performs both RNA editing and DNA editing in normal small intestine. Given the established physiological function of APOBEC1 in editing *APOB* mRNA, it also leads to the conjecture that either APOBEC1 has multiple physiological functions, some mediated by RNA editing and others by DNA editing, or that the DNA editing leading to SBS2/13 is simply collateral damage arising as a result of the high levels of APOBEC1 required to serve its role in *APOB* mRNA editing. “

We have also employed computational simulations to characterise stem cell dynamics (line 75-78), used crypts from individuals who had received chemotherapy to show crypt sweep times (line 119-124), and scrutinized the data further to provide additional insights into the effects of coeliac disease (line 253-265).

Point-by-point response to Reviewers' comments:

Reviewer #1:

Remarks to the Author:

In the manuscript “APOBEC mutagenesis is a common process in normal human small intestine”, Wang et al. perform whole genome sequencing of 342 human small intestinal crypts. They find that the mutation rate does not much differ from the rate observed in the colon, excluding differential mutation rates as a potential reason for the diverging cancer incidence in the small vs. the large intestine. Mutational signatures in general also behave similarly as in the colon, except there is a larger frequency of crypts that have experienced APOBEC mutagenesis (SBS2/SBS13) in the small intestine.

This paper is very well written and exceedingly logically structured, making it a joy to read. The underlying data set of 342 crypt genomes is also impressive. However, I regret to say that I was not able to identify any substantial novel results in the manuscript. The authors apply what are now standard methods (most of them developed by their own group): crypt microdissection and sequencing, calling of somatic variants, mutational signature extraction, mutation rate calculation, phylogenetic reconstruction, and come up with a small portfolio of observations. Methodologically, there is little to nothing to criticize – there is no doubt that the methods are robust, and the research is very well executed. However, the findings are very limited.

Please see the above introductory paragraphs of these responses to help address this criticism.

Specifically,

1) It is interesting to read that the mutation rate in the small intestine does not really deviate from the colon. The authors might have hypothesized at the beginning of the study that they would find lower mutation rates in the small intestine, perhaps partially explaining the reduced cancer risk. However, this is not the case. I think this result is important, but it is essentially negative and also not entirely novel as the authors have already published this result in one of their prior papers (Moore et al., Nature 2021), albeit backed by a smaller data set.

We agree with the reviewer that the lower cancer risk in the small intestine compared to the large intestine is an important long-standing observation that raises fundamental questions about cancer development, and for this reason highlighted it in the text of the manuscript. We agree that, unfortunately, in this study we have not provided further clear insight into why this should be the case. It remains a conundrum.

2) The patchy nature of APOBEC mutagenesis has also been previously published by the same group (Lawson et al., Science 2020). I acknowledge it is a different organ now, but conceptually there appears to be no difference, and the authors do not appear to be any closer to being able to explain the mechanisms underlying these observations.

As the reviewer indicates APOBEC mutagenesis is confined to some (but not all) cells in the bladder epithelium and is similarly so in the small intestine. However, the highly organised nature of the small intestine epithelium with its arrays of morphologically clearly defined crypts, each of which originates

from a single stem cell, allowed us to investigate more definitively whether adjacent clusters of cells/crypts are affected (as predicted if the stimulus is an externally originating but local diffusable stimulus) or whether SBS2/13 mutations are found in individual crypts (with their immediate neighbours lacking them) and the stimulus is therefore likely intrinsic. The latter appears to be the case, a similar conclusion to that drawn from the bladder study but more clearly defined because of the organised microstructure of the small intestine and extent of the data here.

3) The results on kataegis in Figure 4 are also not very surprising given prior knowledge.

We were not sure, in advance of analysing the data, that we would find kataegis, particularly given the lack of genome rearrangement in normal small intestine epithelium (although we agree kataegis was found in normal bladder). We would also not necessarily have predicted the presence of kataegis now given the likely role of APOBEC1 in generating the SBS2/13 mutations. However, it is certainly there and we believe an important point to make in describing the full nature of APOBEC mutagenesis in normal tissues generally and in APOBEC1 mutagenesis specifically.

Other than these (fairly severe) concerns about novelty, I find the paper very polished and have no technical concerns. One minor thought was that it would be helpful if the authors could clarify the anatomical relationship between dissected crypts from the beginning (were they all adjacent and from the same tissue section or were some of them spatially separated?). Figure 3 shows adjacent crypts, but I was not sure whether that applied to all the microdissected specimens.

Some dissected crypts were from the same section and adjacent, and we made efforts to enrich for such sets, but some were from different sections, some were separated but in the same section and, of course, many were from different individuals and biopsies.

We appreciate this suggestion and have added a sentence explaining our sampling strategy at the beginning line 68-70: "From each biopsy, we aimed to collect both spatially adjacent and separated crypts whenever possible."

Reviewer #2:

Remarks to the Author:

This paper reproduces the considerable technological achievement of Lee Six et al, 2019 by undertaking 25X WGS on laser captured normal appearing crypts from the gut. This time the small intestine is assessed and undergoes a very similar analysis pipeline to that reported in the colon, including

construction of phylogenetic trees, comparison with tissue specific cancers and analysis of mutation signatures (indicating a higher SB frequency of characteristic APOBEC mutations).

Although the technical achievement of producing WGS data from such a small amount of human tissue should not be underestimated, the reproduction of the technique and the analysis pipeline from the previous publication inevitably impacts the novelty of the work. Some interesting observations are made, but these are not explored mechanistically or at depth and the paper, somewhat frustratingly, leaves the reader with more questions than it answers.

Please see the introductory paragraphs above of these responses to help address this criticism.

Specific comments.

1. Mutations characteristic for APOBEC are seen at higher frequency in the small intestine than in the colon but this interesting phenomenon is not investigated in any depth. APOBEC1 has tissue restricted expression in the small intestine (Hirano et al, 1997). Is there any similar differential expression of APOBEC3A expression along the length of the human gut?

We sincerely appreciate this suggestion and prompt from the Reviewer which has helped us substantially. At the time of the initial submission, we briefly checked that APOBEC3A/3B expression is not higher in small intestine compared to in large intestine, but did not include it in the original manuscript. However, during revision and in response to Referee's comments we carried out a more comprehensive comparison of APOBEC gene expression across multiple datasets, generating the hypothesis that we believe can explain the difference in SBS2/13 levels mechanistically.

In short, our finding is that APOBEC1 is the only APOBEC family enzyme that is consistently highly expressed in small intestine compared to large intestine epithelium across all publicly available datasets that we have investigated (~10 to 20-fold higher in epithelium from bulk RNA seq data, over 20 to 40-fold in stem cells and transit amplifying cells from single cell RNA seq data). The sequence context of the APOBEC C>N DNA editing mutations in small intestine is also consistent with the sequence context of APOBEC1 C>N DNA editing generated in experimental systems (while not excluding APOBEC3A).

Given the known physiological function of APOBEC1 of mediating C-to-U editing on *APOB* mRNA, we speculate that APOBEC DNA editing in the small intestine may simply be "collateral damage" as a result of high levels of APOBEC1, rather than having a specific function.

We have modified the manuscript to add analysis and discussion around expression data accordingly. These changes are cited above in the introduction to our response to Reviewers.

2. Can mutation burden/phylogenetic trees be used as substrate for models to infer human small intestinal stem cell dynamics or clonal expansion and contrast with that in the colonic data of Lee Six et al? (as per work of Doug Winton, Ben Simons)

We have further investigated this problem during manuscript revision. We can estimate a time to the most recent common ancestor (MRCA) stem cell of the crypt through approximate Bayesian computation, which is to simulate the stem cell dynamics in a crypt, and then compare the simulated final variant allele frequency distribution to the real data to pick the best simulation for parameter estimation. The average time to MRCA estimated from the best 1% simulation ranges from 0.8-4.6 yrs across individuals, with a median=2.6. Therefore, we can say that it only takes a few years at most for the MRCA to occupy the crypt, not decades.

Although this time appears to be shorter compared to the 5.5 yrs (95% confidence interval, 1–10.5) estimated from the colonic data, because the confidence intervals overlap, we do not believe that it is appropriate to conclude that time to MRCA is shorter in the small intestine from this dataset. However, the possibility is not excluded, and it would be worthwhile to set up follow-up experimental studies to directly measure this.

It would be not possible to decouple the number of stem cells and stem cell replacement rate without support from experimental evidence (similar work conducted in the small intestine compared to the work the Reviewer has mentioned in the colon), therefore we hesitate to make any comment on them.

3. Following on from this - Mutation rates are found to be comparable to human colon, but was this normalised for putative differences in colonic and small intestinal crypt stem and progenitor cell number?.

The mutation rate was not adjusted for stem cell/progenitor cell number. Each crypt is a clone derived from a single stem cell existing at some point in the past and we are only able to detect clonal mutations

using the applied sequencing approaches. Thus we detect the number of mutations in that most recent common ancestor of each crypt. As the Referee is suggesting, this could affect mutation rate calculations due to different sweep times between colon and small intestine (and thus different intercepts), but we used the slopes of the lines of mutation accumulation which are not dependent on the intercepts and which therefore should not be affected. It turns out, anyway, from other studies and additional analysis here that these sweep times are relatively small (a small number of years) and therefore would not substantially affect mutation rate calculations.

4. A post chemo patient had a much higher mutation rate and was excluded from analysis, which feels like an opportunity lost. Why not further explore this interesting observation to draw inferences about cell dynamics/turnover in the gut following cytotoxic drug therapy or following radiation enteritis.

We thank the Reviewer for this suggestion. We apologize that we should have made it clearer in the initial submission that this individual has been included in the phylogenetic analyses. The samples from this individual were excluded only from the mutation rate estimation because of the large numbers of mutations attributed to chemotherapy-related signatures which would distort estimates of normal mutation rates.

We have shown the phylogenetic trees of the post chemotherapy patients in current Extended Data Fig. 3 (this figure has now been edited to show which patients are post-chemo), and Variant Allele Frequency plots showing clonality in Extended Data Fig.1. The post-chemotherapy patients do indeed bring some interesting insights. In particular, an interesting finding is how quickly chemotherapy-related signatures start to appear. In PD28690, 5-6 weeks of oxaliplatin treatment before death was sufficient for some cells in small intestinal crypts to accumulate a few hundred chemotherapy-induced mutations (with bimodal VAF distributions). In PD43853, 5 months after the first cycle of fluorouracil and oxaliplatin treatment, cells carrying chemotherapy-induced mutations had already occupied whole crypts (even when restricted to monoclonal mutations with VAF > 0.4, we can still find considerable amounts of mutations from chemotherapy signatures SBS17b and SBS35).

Even in PD42835, who only received 3 weeks of Irinotecan and 5-FU treatment before the biopsy was collected, a small amount of SBS17b (the 5-FU signature) was detected in HDP *de novo* signature extraction (which is below the final 5% threshold and not shown on the tree) and small peaks around zero in VAF distributions, indicating that subclones with 5-FU induced mutations have emerged.

Example spectra of monoclonal mutations (VAF>0.4) in PD43853, 5 months after the first cycle of fluorouracil and oxaliplatin treatment, show strong signals of chemotherapy signatures SBS17b and SBS35:

5. It is mentioned that coeliac didn't impact mutation burden in phenotypically normal crypts, but we were told very little about the disease state. Was there characteristic crypt hyperplasia seen in these patients or had this resolved post treatment?.

We thank the Reviewer for raising this question.

To facilitate the understanding of the disease state, we have also added a sheet (coeliac metadata) in Extended Data Table 5 to include metadata for coeliac disease state to the extent that we can access it. Key information such as duration of symptoms, whether the coeliac disease is refractory or not, and whether they were on a gluten-free diet were included. With respect to the second question, we can see the characteristic features of coeliac disease under the microscope.

Images of small intestine biopsies from coeliac patients:

PD46566

PD46563

PD46568

PD46562

PD46565

PD46573

A normal example(PD45773b), showing tall villi and shallow crypts:

Although the impact of a coeliac history on the total SBS mutation burden is not significant, further examination of the data revealed it has a minor impact on the ID burdens: adding indel burden by 1.6 IDs per year (95% confidence interval 0.3-2.9). Also, further investigation into mutational burdens for individual signatures identified that a coeliac history leads to an increased burden in SBS1 (4.8 mutations per year, 95% confidence interval 0.4-9.2), but not other signatures. We added Extended Data Fig.8 to demonstrate the disease effect and edited line 255-265:

“Although the number of affected individuals is small and estimates correspondingly uncertain, coeliac disease increased the SBS1 burden by 4.8 mutations per year (95% confidence interval 0.4-9.2, $P = 0.034$ by t-test), and indel burden by 1.6 IDs per year (95% confidence interval 0.3-2.9, $P = 0.017$ by t-test), indicative of 1.43-fold increased small indel and 1.24-fold increased SBS1 mutation rates (in these analyses we used age to estimate mutation rates and therefore the mutation rates during active disease periods could be considerably higher) (**Extended Data Fig. 8**). However, the total mutation rate, driver mutation rate, complement of mutational signatures and rates of mutational signatures other than SBS1 were not detectably affected. The SBS1 mutation rate has previously been correlated with cell division rates²², suggesting that there is accelerated crypt stem cell proliferation in coeliac disease.”

6. Is stable mutation rate and lack of clonal expansion also the case in other small intestinal conditions such as Crohns disease?.

Olafsson *et al.* (Cell 2020) have performed a detailed investigation into the mutational landscape and clonal dynamics of Inflammatory Bowel Disease-affected colon, including Crohn's disease. They did find an accelerated mutation rate and evidence of clonal expansion in the affected colonic crypts. Unfortunately, they did not cut and sequence any small intestinal crypts from the patients with Crohn's disease, and neither do we have such data at hand. This would be an interesting point to look at in the future when these data are available.

7. The difference in cancer prevalence between the small bowel and colon is mentioned in passing but no real attempt is made to use these data and that available from Lee Six et al, 2019 to explore this longstanding and very important question.

We explored several key aspects that we could confidently infer including mutation rate, driver mutation prevalence and the frequency of structural variants. In the revised manuscript we also checked the stem cell dynamics and estimated a time to the most recent common ancestor for each normal adult individual.

Unfortunately, none of these factors effectively explain the substantial difference in cancer incidence rate between small and large intestine, and we have been open about this. Nevertheless, the results indicate that the reason for the lower cancer risk in the small intestine does not lie in the mutation burdens of adult epithelial cells (a proposal that was often made in the past), but in other factors, speculatively, for example, epigenetic changes and other potential effects of the presence of the microbiome in the colon.

In summary, the paper replicates the considerable technological achievement of previous work in a different organ, but fails to truly exploit the capacity to assess normal crypt mutation burden to further or mechanistically explore some of the interesting observations that are made.

Reviewer #3:

Remarks to the Author:

Manuscript describes a study of whole-genome spectra of changes in over 300 individual small intestinal crypts obtained by microdissection from tissue biopsies of 39 individuals. Each microdissection collected

clonal group of cells which allows sufficient accuracy of calling somatic genome changes present in a cell that gave rise to a crypt-clone. Most frequent changes were single base substitutions (SBS). Agnostic signature extraction as well as comparison with previously identified SBSes revealed several mutational signatures of unknown origin, which could be caused by either endogenous or by exogenous DNA lesions. They also found two well defined endogenous mutational processes, clock-like SBS1 stemming from deamination of 5me-C in CpG motifs and a signature of cytidine deamination by single-stranded DNA-specific APOBEC cytidine deaminases. The latter signature is one of the most abundant in human cancers, however, was rarely found in normal somatic cells (so far on bladder and bronchial epithelia). Analysis of extended APOBEC mutational context indicated that the APOBEC3A is the likely source of mutations. The presence of APOBEC mutagenesis was supported by observation of mutation clusters highly enriched with APOBEC mutational motifs. Detailed analysis of adjacent crypts established that APOBEC mutagenesis is not uniformly distributed across small intestinal epithelium but is rather episodic. The conclusions about the presence and the features of APOBEC mutagenesis are of interest to the fields of genome stability, mutagenesis, and cancer biology.

Note:

Since pages in the Manuscript file were not numbered positions in the manuscripts are referenced by quotations.

Critique.

1. Insufficient source data and description of analysis do not allow recapitulation and reuse of results, and validation of the conclusions.

A. Insufficient source data on mutation call statistics and signatures in individual samples.

In this study, multiple samples (crypts) were sequenced from each individual. Supplementary Table 4 lists metadata for individuals, but not for sequenced crypts. However, all display items supporting conclusions of the manuscript contain information derived from sequencing genomes of individual samples/crypts. The Supplementary Table listing all sequenced crypts, their anatomical origin, numbers of SBS, SV, ID, CNV for each sequenced sample as well as SBS signature mutation load determined for each crypt is required. It took a while from me to locate such a Table on https://github.com/YichenWang1/small_bowel/blob/main/data/stat_summary.txt [github.com], and I do not see any reason for making any interested reader to go through this search instead of just placing this source data for easy access into Supplementary part. I attached Excel version of this Table "Crypts_stat_summary.txt.xlsx" for convenience of all parties involved in reviewing or/and handling this manuscript. Authors should also provide Readme explaining the content of each column and refer in Figure legends and/or in Methods to the columns containing values used for creating each display item. So far there are some inconsistencies between this Table and Figures or/and Method description. The

inconsistencies found by this reviewer are in Specific Points section.

We would like to thank the reviewer for the valuable feedback on facilitating the reuse of the data. We have now moved the summary file into the Supplementary (Extended_Data_Table3_crypt_summary.csv) as suggested. Github repository has also been updated accordingly, explaining the analysis workflow as well as corresponding location of code and input data. The workflow and code in R Markdown for creating the figures related to this table can also be found at:

https://github.com/YichenWang1/small_bowel/tree/main/Mutation_burden

We hope this will help create a better user experience, and we would be happy to modify/add any data and information in the future, if anyone spots anything we have missed.

We also thank the reviewer for noting the inconsistency. We recognise that the original text could have caused confusion and difficulty in replication (as some of the descriptions were not put into the manuscript, but in the code and reporting summary instead). Therefore, we have edited the text in the Methods as below to include all necessary details for replicating the results in line 575-582:

“When modelling mutation rates, samples were excluded with <15-fold coverage and from one individual (PD43853) with a substantial number of chemotherapy induced mutations. In addition, two biopsies (PD43851j_P52_DDM_E2, PD46565c_lo0009) have a mutational landscape dominated by SBS5/40, which is distinct from the mutational landscape of normal small intestinal crypts, and similar to that of Brunner's glands, despite their crypt-like appearance under microscopy inspection. These two samples were kept and reported for the comprehensiveness and transparency of this dataset, but not included in the statistical modelling of mutation rates (leaving 306 crypts).”

B. Lack of mutation call information used as direct input into signature and mutation cluster analyses.

Github folder https://github.com/YichenWang1/small_bowel/tree/main/data/vcf [github.com] contains mutation calls for each sequenced sample/crypt. However, in Methods they state:

“Before running HDP [signature extraction pipeline], we assigned mutations to branches on the phylogenetic tree. This avoided counting the same mutation within one patient multiple times. To avoid

overfitting, we only kept branches with > 50 mutations as input.”

This implies that mutations in the trunk of a phylogenetic tree that, based on Extended Data Figure 4, were present in up to four individual samples were assigned to a single sample from a group joined by common phylogeny. Sample specific mutation catalogues in VCF or MAF format adjusted by removal of duplicated calls should be provided as a source data to allow recapitulation and reuse of results, and validation of the conclusions. While authors provide scripts that were used for analysis, it would require unreasonable effort from interested colleagues to recreate deduplicated VCF or MAF catalogues. Moreover, it would not be possible to generate the same deduplicated catalogues, because assignment of the trunk to a single catalogue was made by authors based on an undisclosed rule or just randomly.

We thank the reviewer for pointing this out and apologise for the inconvenience caused. Now the VCF files for individual crypts are publicly available at our Github repository (with the location described in Readme file as well): https://github.com/YichenWang1/small_bowel/tree/main/data/vcf . We acknowledge how valuable these series of suggestions are for building a community with better data-sharing practice, and would like to thank the reviewer for prompting us on this.

2. Arbitrary threshold of 5% for APOBEC SBS2/13 as well as other SBS signatures for designating a sample as a “signature positive” (Methods sections “Mutational signature extraction...” and “Comparison with the small bowel adenocarcinoma...”

Note that percentage would depend not only on the number of APOBEC motif mutations in C (G) but also on other C (G)-motifs as well as on the number of T (A). Authors may want to try enrichment statistically evaluated for the excess of APOBEC motif C (G) mutations over the presence of APOBEC motifs in surrounding genomic background. For the enrichment with APOBEC mutational motif a rigorous statistical evaluation does exist (refs 27, 44, 45 and <https://github.com/NIEHS/P-MACD> [github.com]). This method can also detect APOBEC mutational signature in clusters where it is more pronounced than in whole-genome catalogue.

We have tested the suggested tool for the enrichment of the APOBEC mutational motif with default parameters and motif as an independent verification.

Using this tool, TCW mutations were found to be significantly enriched ($P < 0.05$ from Fisher’s exact test with Benjamini-Hochberg correction) in 37 out of 417 (8.9%) phylogenetic branches in small intestine and 3 out of 1066 (0.3%) branches in colorectum ($P = 2.2 \times 10^{-19}$, Chi-squared test).

In our previous report, SBS2/13 were found in 68 out of 417 (16.3%) phylogenetic branches in small intestine, ~28-fold more frequently than the 6 out of 1075 (0.6%) branches in colorectum ($P = 1.6 \times 10^{-35}$, Chi-squared test). Therefore, the relatively ratio and conclusion remains largely the same. We could only obtain the mutation matrices for small bowel adenocarcinomas, therefore were not able to test P-MACD on the cancer data (which needs MAF files as input).

With respect to the 5% threshold of signature detection, we concede that it is arbitrary, and therefore may introduce false positives and false negatives. Nevertheless, this was used to maintain consistency in signature attribution of the other signatures (which could not rely on tools like P-MACD to test a certain motif), and we have manually checked the raw mutational spectrums to make sure the signature attribution results are sensible. We found with higher thresholds such as 10% we would fail to report signals clearly visually present in the mutational spectrum of a number of cases.

3. Cluster analysis

A. Information about the content of deduplicated sample-specific mutation catalogues is essential for mutation cluster (kataegis) calling. The Methods section “Identifying kataegis” does not describe how authors avoided using several times the calls belonging to the trunk of a phylogeny tree. This should be clarified in methods with reference to deduplicated mutation catalogues.

Since the algorithm runs on an individual vcf file, it cannot automatically remove duplicated mutations. Therefore, we have checked the output and labelled the duplicated mutation on the shared trunk (shared mutations will have multiple labels in the ‘sample’ column), and provide the results in Extended_Data_Table4_kataegis.csv. We have now added the sentence to the Method description of ‘Identifying kataegis’ line 631-632:

“Output results were manually inspected to avoid counting shared mutation clusters multiple times.”

B. What was the minimal number of mutations in a group with less than 10 kb inter-mutational distance that was evaluated as a potential cluster? References 27, 44, 45 as well as PMID 31568516 evaluated groups of at least 2 mutations. Strand-coordinated clusters with as little as 2 mutations showed enrichment with APOBEC mutational motif greater than scattered mutations in the same genomes.

We did not set a hard threshold on the minimal number, but with an adjusted P-value $< 10^{-4}$ after

Bonferroni correction, our mutation clusters consist of at least 3 mutations. This threshold is stringent as we wish to emphasise the elimination of false positives when reporting *kataegis* in a normal tissue and supporting the presence of APOBEC mutagenesis.

C. There are several samples with APOBEC mutation clusters, but still APOBEC-negative genome-wide. Presence of APOBEC mutation cluster(s) is the evidence of APOBEC mutagenesis operating in the history of a sequenced genome. The lack of APOBEC mutagenesis detectable on a genome-wide scale may be due to low abundance of single-stranded DNA accessible to APOBEC enzymes. Presence of APOBEC clusters in the genome of each sample should be indicated on the Supplementary Table exemplified by “Crypts_stat_summary.txt.xlsx” and on illustrations of phylogeny trees. It is important to know if APOBEC clusters would be detected in the trunks and/or in the branches which are qualified as APOBEC negative.

We have added columns for the number of *kataegis* foci, and for the total number of *kataegis* mutations in each crypt to the Extended_Data_Table3_crypt_summary.csv as suggested. We also have added the number of APOBEC clusters on the branch of each phylogenetic tree (Extended Data Fig.3). Indeed, there are examples where APOBEC clusters were detected on the APOBEC-negative branches, possibly indicating insufficient ssDNA substrate at the time of APOBEC activation.

Specific points.

1. “When modelling mutation rates, only samples with >15-fold coverage were included, and one patient (PD43853) with a substantial number of mutations due to chemotherapy was excluded (n = 306)”

Blacklisted samples should be specified in the Supplementary Table exemplified by “Crypts_stat_summary.txt.xlsx”. So far I was unable to get 306 whitelisted samples applying filters quoted above.

We hope our response to Critique A.1 have addressed this concern, and we have added detailed description in the Methods section line 575-582 about how to get the 306 samples:

“When modelling mutation rates, samples were excluded with <15-fold coverage and from one individual (PD43853) with a substantial number of chemotherapy induced mutations. In addition, two biopsies (PD43851j_P52_DDM_E2, PD46565c_lo0009) have a mutational landscape dominant by SBS5/40 and with lower mutation burden, which is distinct from the mutational landscape of normal

small intestinal crypts, and similar to that of Brunner's glands, despite their crypt-like appearance under microscopy inspection. These two samples were kept and reported for the comprehensiveness and transparency of this dataset, but not included in the statistical modelling of mutation rates (leaving 306 crypts).”

We also included the workflow and code about this analysis at:

https://github.com/YichenWang1/small_bowel/tree/main/Mutational_burden

2. “commonly found in many cancer types... and uterus (refs1,2)”.

Authors may want to add another reference #45 (2013) here, which used an alternative to NMF motif-specific approach to demonstrate ubiquitous presence of APOBEC mutagenesis in a number of human cancer types.

Thank you for the Reviewer’s indication of this paper, we have added this reference here.

3. “However, the median burden of APOBEC mutations in normal small intestine was ~7-fold lower than that observed in small bowel adenocarcinoma”

Replace "the median burden of APOBEC mutations " with "SBS2/13 burden" to be consistent with Extended Data Fig. 6 legend and to clarify that was it the contribution of SBS2+SBS13 obtained as an outcome of HDP, not the simple counts of APOBEC motif mutations. This is important to allow analyses in follow-up studies to be set consistent with this work.

We thank the Reviewer for this comment and would be happy to modify this as suggested.

4. “Enrichment of SBS2/13 mutations with a pyrimidine rather than a purine (ref 27)...”

Note that only peaks with A in -2 positions are labelled on Extended Data Fig 4; Peaks with another purine– G are not labeled, which makes it confusing to a reader

What values are on y-axis of Extended Data Fig 4? Is the statement of higher presence of pyrimidines over purines supported by statistics? Compare with the estimates obtained with the algorithm described in ref 27 and available at <https://github.com/NIEHS/P-MACD> [github.com] . This algorithm generates

p-values.

What is the preference in -2 position for mutations in APOBEC mutation clusters? Based on Ref 27 APOBEC3B may operate in all genomes, while more potent APOBEC3A mutagenesis may mask the impact of APOBEC3B. Note that APOBEC3B mutagenesis is readily detectable in mutation clusters of cancer genomes with low level of APOBEC mutagenesis (PMID 30537516).

We thank the Reviewer for their suggestions. We have modified the Extended Data Figure to add y-axis values and have labelled all the peaks. To facilitate the comparison, we also have added a Panel b to show the sequence logo of the extracted signatures, and have used the suggested tool P-MACD to obtain enrichment scores (shown in current Extended Data Fig. 7 Panel c) and calculated P-values based on Fisher's exact test. The results consistently showed a depletion of the RTCA motifs preferred by APOBEC3B, but could not discriminate between APOBEC3A and APOBEC1 (also acting on TCN motifs on DNA) as the operating enzymes.

5. "SBS2/13 mutations in cancer are found in localised, high-density clusters referred to as 'kataegis' (refs 29,30)"

Reference 44 - Roberts 2012 - also demonstrated strand coordinated-clusters with clear APOBEC mutational motif and was published simultaneously with reference 29.

Thanks for reminding us of this, we have added this citation here.

6. Replace ref 18 from preprint to published paper.

We have updated our reference list accordingly.

7. "For identified SBS signatures, signatures with ≥ 0.9 cosine similarity with the reference were considered as the same signatures"

SBS2+SBS13 should be used for cosign similarity with extracted HDP signatures here and in the analyses shown in Extended Data Fig. 8j. In this figure, Component 4, which is clearly APOBEC signature has cosign similarities approx. 0.4 with either SBS2 or SBS13, however it is clear that the total of SBS2+SBS13 will show very high cosign similarity with Component 4.

We understand the Reviewer's concern about preserving Component 4 to be SBS2 and SBS13 as a whole by using this synthetic reference 'SBS2+SBS13'.

In our workflow, it also ended up that Component 4 was decomposed into SBS2 and 13, but not any other signatures. We agree with the Reviewer that it would be clearer to compare Component 4 with SBS2+SBS13 directly, and we have modified panel j of our Extended Data Figure accordingly as below:

The workflow and results of the decomposition can be checked at:

https://github.com/YichenWang1/small_bowel/blob/main/Signatures/Signatures.md

8. Extended Data Fig. 4.

Color codes unreadable or/and indistinguishable. Increase size (possibly by making two panels) and put the SBS name on top of each section of the corresponding branch. Phylogeny trees support important conclusions of this manuscript, therefore source data with numeric values used to build this figure should be provided.

We have increased the figure size as suggested, and grouped the trees by features (coeliac, post-chemotherapy, children, all others) to make them more readable. We have also uploaded the source file in .tree format, as well as a more readable csv format to our Github repository:

https://github.com/YichenWang1/small_bowel/tree/main/data/phylogenetic_trees

Decision Letter, Appeal

6th Oct 2022

Dear Mike,

How are you? I hope that you are well.

Please forgive the delay in returning this decision to you. Thank you so much for asking us to reconsider our decision on your manuscript "APOBEC mutagenesis is a common process in normal human small intestine". I have now discussed the points of your appeal with my colleagues, and we would like to send the revised manuscript back out for review.

This stage of the appeal necessitates a resubmission of all your materials - I'm sorry about this additional step.

When preparing a revision, please ensure that it fully complies with our editorial requirements for format and style; details can be found in the Guide to Authors on our website (<http://www.nature.com/ng/>).

Please be sure that your manuscript is accompanied by a separate letter detailing the changes you have made and your response to the points raised. At this stage we will need you to upload:

1) a copy of the manuscript in MS Word .docx format.

2) The Editorial Policy Checklist:

<https://www.nature.com/documents/nr-editorial-policy-checklist.pdf>

3) The Reporting Summary:

(Here you can read about the role of the Reporting Summary in reproducible science:

<https://www.nature.com/news/announcement-towards-greater-reproducibility-for-life-sciences-research-in-nature-1.22062>)

Please use the link below to be taken directly to the site and view and revise your manuscript:

[redacted]

I look forward to receiving everything.

Kind regards,

Safia

Safia Danovi
Editor
Nature Genetics

Decision Letter, first revision:

Our ref: NG-LE60027R1

1st Nov 2022

Dear Dr. Stratton,

How are you? I hope that you are well.

Thank you for submitting your revised manuscript "APOBEC mutagenesis is a common process in normal human small intestine" (NG-LE60027R1). It has now been seen by the original referees and their comments are below.

We discussed Reviewer #2's request for experimental validation extensively and we agree that these experiments would significantly fortify the paper. However, we also appreciate that these experiments are not trivial and might not necessarily lead to a straightforward answer. As such, we are prepared to overrule this request as we are comfortable with the overall novelty afforded by the study as it is. Considering this together with the support for publication voiced by Reviewers #1 and #3, we'll be happy in principle to publish your paper in Nature Genetics, pending minor revisions to satisfy our editorial and formatting guidelines.

Sincerely,

Safia

Safia Danovi
Editor
Nature Genetics

Reviewer #1 (Remarks to the Author):

I thank the authors for their answers to my questions. During the first round of review, my main concern was novelty; I had no technical concerns and found the paper to be well-written and informative. In order to strengthen the novelty component, the authors have in this revised version added a bioinformatic analysis of gene expression in small vs. large intestine, generating the hypothesis that increased SBS2/13 mutation rates in the small intestine are caused by high expression

of APOBEC1. I like this analysis, but I am afraid that it does relatively little to add substance or novelty to the paper. It generates a hypothesis about the generally higher SBS2/13 mutation burden in the small intestine, but in the absence of further corroboration, this circumstantial evidence does not advance the narrative very much, in my opinion. Importantly, constitutively high APOBEC1 expression cannot explain the observed episodic/patchy nature of SBS2/13 mutagenesis (probably the most intriguing observation made by the authors, also in the context of other tissues). That being said, I continue to think that this paper is based on a very valuable data set and is crafted extremely well, so I think a good case could be made for publication in Nature Genetics if editorial interest in the topic is high.

Reviewer #2 (Remarks to the Author):

This manuscript is improved on the last iteration. I am reasonably happy with the response to many of my points, however my fundamental concern over the descriptive nature of the observations and the novelty has not been fully addressed.

The new stated hypothesis that tissue-restricted APOBEC1 activity generates DNA damage to contribute to the SBS2/13 signature exclusively in the small intestine is interesting. The idea that this is collateral to its known RNA splicing function to aid lipid absorption is intriguing and starts to get to a mechanistically satisfying answer as to why these mutation levels and signatures are seen in this distinct tissue.

However this hypothesis is currently speculative, and unproven. The papers quoted as showing that APOBEC1 generates a SBS2/13 signature are from E. Coli and yeast cells respectively. The proposed experiments in the discussion of the paper to look at mutation burden in human SB organoids should be achievable to show human tissue relevance. As there is some literature evidence to show that p53 activation (through Adriamycin) can enhance APOBEC1 activation status (PMID 20890106), there may be some opportunity to speed up the process by looking at before and after treatment tissue samples.

Minor point

line 248. Spelling APOBEC3A

Reviewer #3 (Remarks to the Author):

Authors adequately addressed my comments and added a new interesting angle on potential role of APOBEC1 in mutagenesis. The manuscript is now suitable for publication in Nature Genetics.

Author Rebuttal, first revision:

Reviewer #1:

Remarks to the Author:

I thank the authors for their answers to my questions. During the first round of review, my main concern was novelty; I had no technical concerns and found the

paper to be well-written and informative. In order to strengthen the novelty component, the authors have in this revised version added a bioinformatic analysis of gene expression in small vs. large intestine, generating the hypothesis that increased SBS2/13 mutation rates in the small intestine are caused by high expression of APOBEC1. I like this analysis, but I am afraid that it does relatively little to add substance or novelty to the paper. It generates a hypothesis about the generally higher SBS2/13 mutation burden in the small intestine, but in the absence of further corroboration, this circumstantial evidence does not advance the narrative very much, in my opinion. Importantly, constitutively high APOBEC1 expression cannot explain the observed episodic/patchy nature of SBS2/13 mutagenesis (probably the most intriguing observation made by the authors, also in the context of other tissues). That being said, I continue to think that this paper is based on a very valuable data set and is crafted extremely well, so I think a good case could be made for publication in Nature Genetics if editorial interest in the topic is high.

We thank the Reviewer for the recognition.

The hypothesis of APOBEC1 as the responsible enzymes is speculative for now. But as we propose this to the field in this study, we believe this would be an invitation to future experiments and validations. We look forward to seeing evidence around this generated by us and possibly others in the field who are interested.

The constitutive expression versus the observation of very few episodes in life is interesting by itself, and would indicate a constitutive protective mechanism against APOBEC mutagenesis in normal tissues, eg. lack of substrates from double strand break at the time of APOBEC expression.

Reviewer #2:

Remarks to the Author:

This manuscript is improved on the last iteration. I am reasonably happy with the response to many of my points, however my fundamental concern over the descriptive nature of the observations and the novelty has not been fully addressed.

The new stated hypothesis that tissue-restricted APOBEC1 activity generates DNA damage to contribute to the SBS2/13 signature exclusively in the small intestine is interesting. The idea that this is collateral to its known RNA splicing function to aid lipid absorption is intriguing and starts to get to a mechanistically satisfying answer as to why these mutation levels and signatures are seen in this distinct tissue.

However this hypothesis is currently speculative, and unproven. The papers quoted as showing that APOBEC1 generates a SBS2/13 signature are from E. Coli and yeast cells respectively. The proposed experiments in the discussion of the paper to look at mutation burden in human SB organoids should be achievable to show human tissue relevance. As there is some literature evidence to show that p53 activation (through Adriamycin) can enhance APOBEC1 activation status (PMID 20890106), there may be some opportunity to speed up the process by looking at before and after treatment tissue samples.

We thank the reviewer for the suggestions on the experiments, and we agree that these experiments would be helpful in demystify this speculation. Although these experiments are not trivial and therefore we may not be included them for now, we will consider this design carefully for a study in the future.

Minor point

line 248. Spelling APOBEC3A

Thanks! This typo has been corrected.

Reviewer #3:

Remarks to the Author:

Authors adequately addressed my comments and added a new interesting angle on potential role of APOBEC1 in mutagenesis. The manuscript is now suitable for publication in Nature Genetics.

We thank the Reviewer for the kind support

Final Decision Letter:

16th Dec 2022

Dear Dr. Stratton,

I am delighted to say that your manuscript "APOBEC mutagenesis is a common process in normal human small intestine" has been accepted for publication in an upcoming issue of Nature Genetics.

Your paper will be published online after we receive your corrections and will appear in print in the next available issue. You can find out your date of online publication by contacting the Nature Press Office (press@nature.com) after sending your e-proof corrections. Now is the time to inform your Public Relations or Press Office about your paper, as they might be interested in promoting its

publication. This will allow them time to prepare an accurate and satisfactory press release. Include your manuscript tracking number (NG-A60027R2) and the name of the journal, which they will need when they contact our Press Office.

Please note that *Nature Genetics* is a Transformative Journal (TJ). Authors may publish their research with us through the traditional subscription access route or make their paper immediately open access through payment of an article-processing charge (APC). Authors will not be required to make a final decision about access to their article until it has been accepted. [Find out more about Transformative Journals](https://www.springernature.com/gp/open-research/transformative-journals)

Authors may need to take specific actions to achieve [compliance with funder and institutional open access mandates](https://www.springernature.com/gp/open-research/funding/policy-compliance-facts). If your research is supported by a funder that requires immediate open access (e.g. according to [Plan S principles](https://www.springernature.com/gp/open-research/plan-s-compliance)) then you should select the gold OA route, and we will direct you to the compliant route where possible. For authors selecting the subscription publication route, the journal's standard licensing terms will need to be accepted, including [a](https://www.nature.com/nature-portfolio/editorial-policies/self-archiving-and-license-to-publish). Those licensing terms will supersede any other terms that the author or any third party may assert apply to any version of the manuscript.

Please note that Nature Portfolio offers an immediate open access option only for papers that were first submitted after 1 January, 2021.

To assist our authors in disseminating their research to the broader community, our SharedIt initiative provides you with a unique shareable link that will allow anyone (with or without a subscription) to

read the published article. Recipients of the link with a subscription will also be able to download and print the PDF.

If you have not already done so, we invite you to upload the step-by-step protocols used in this manuscript to the Protocols Exchange, part of our on-line web resource, natureprotocols.com. If you complete the upload by the time you receive your manuscript proofs, we can insert links in your article that lead directly to the protocol details. Your protocol will be made freely available upon publication of your paper. By participating in natureprotocols.com, you are enabling researchers to more readily reproduce or adapt the methodology you use. [Natureprotocols.com](https://natureprotocols.com) is fully searchable, providing your protocols and paper with increased utility and visibility. Please submit your protocol to <https://protocolexchange.researchsquare.com/>. After entering your [nature.com](https://www.nature.com) username and password you will need to enter your manuscript number (NG-A60027R2). Further information can be found at <https://www.nature.com/nature-portfolio/editorial-policies/reporting-standards#protocols>

Sincerely,

Safia Danovi
Editor
Nature Genetics